# Discrete and continuous mechanisms of temporal selection in rapid visual streams

Sébastien Marti[1,2,3] & Stanislas Dehaene[1,2,3,4]

Humans can reliably detect a target picture even when tens of images are flashed every second. Here we use magnetoencephalography to dissect the neural mechanisms underlying the dynamics of temporal selection during a rapid serial visual presentation task. Multivariate decoding algorithms allow us to track the overlapping brain responses induced by each image in a rapid visual stream. The results show that temporal selection involves a sequence of gradual followed by all-or-none stages: (i) all images first undergo the same parallel processing pipeline; (ii) starting around 150 ms, responses to multiple images surrounding the target are continuously amplified in ventral visual areas; (iii) only the images that are subsequently reported elicit late all-or-none activations in visual and parietal areas around 350 ms. Thus, multiple images can cohabit in the brain and undergo efficient parallel processing, but temporal selection also isolates a single one for amplification and report.

[1] NeuroSpin Center, Commissariat à l'Energie Atomique, F-91191 Gif sur Yvette, France. [2] Cognitive Neuroimaging Unit, Institut National de la Santé et de la Recherche Médicale, U992, F-91191 Gif sur Yvette, France. [3] Université Paris-Sud 91405, Orsay, France. [4] Collège de France, F-75005 Paris, France. Correspondence and requests for materials should be addressed to S.M. (email: sebastien.marti@yahoo.fr)

The brain is continuously bombarded by sensory inputs and comprises efficient massively parallel architectures for processing them. Only a subset of the available information is selected in space and time[1–4], and gated to awareness. Such parallel followed by serial processing is exemplified in rapid serial visual presentation (RSVP), an experimental paradigm in which series of stimuli are briefly flashed on a screen (typically ~ 10 per second). Even at such a fast presentation rate, sentence reading and picture recognition remain efficient[5–7]. Nevertheless, under such conditions, the brain lacks the ability to fully process each stimulus: subjective visibility is degraded and some stimuli remain subjectively unperceived[8]. Stimuli also compete with each other for access to higher resources, as evidenced for instance by the fact that, during RSVP, the detection of a target stimulus impedes the perception of a second target during almost half a

second (the Attentional Blink, AB)[9]. In the present study, we explore how the brain selects relevant information from rapidly changing visual inputs.

Recent neuroimaging evidence has started to shed light on how relevant information is processed in RSVP. The identification of a target stimulus involves a complex chain of brain processes that can operate in parallel to another task for > 300 ms[10–12]. Between 350 and 450 ms, relevant stimuli compete for attentional signals arising from posterior parietal cortex[10]. Eventually, the selected stimulus accesses awareness and triggers the synchronized activation of a capacity limited fronto-parietal network at ~ 450 ms[10–13], where information is processed serially.

Although these results give a detailed picture of the sequence of stages involved in the detection of a target stimulus, they leave open the nature of the temporal selection process. Behavioral

**Fig. 1** Experimental design and behavioral evidence for gradual temporal selection. **a** Schematic representation of parallel and serial model of temporal selection. From top to bottom are represented the stream of stimuli, the activity in sensory areas, the gradual selection hypothesis, and the discrete selection hypothesis. Each colored curve represents the predicted brain response for one stimulus in the RSVP at a certain stage of processing. In both models, stimuli are integrated in parallel at the sensory level. In the gradual selection model, attention selects multiple stimuli simultaneously. All stimuli benefit from attentional enhancement but at varying degrees. By contrast, in the discrete selection model, attention is all-or-none and only one stimulus is selected for further processing. **b** Schematic representation of the tasks. In the localizer task (upper part), one stimulus was presented and subjects had to indicate their category. During the dual-task (lower part), stimuli were presented in a rapid visual stream. Subjects were instructed to first determine the color of the very first stimulus (T1) and then identify the image surrounded by a green square (T2). Subjects made systematically three guesses for T2 by order of preference (i.e., from the most probably correct response to the least probably correct). For copyright reasons, the stimuli used in the experiment are replaced by representative images. **c** Distribution of subjects' reports. Rows represent Guess 1, 2, and 3 and columns represent inter-target lags (blue, magenta, red and orange for Lag 1, 3, 7, and 9, respectively). For each position in the RSVP, the proportion of trials was compared to the chance level. A significant difference is indicated by an asterisk (*$P < .05$; **$P < .01$; ***$P < .001$). Error bars represent standard error to the mean. **d** Average (± standard error) mode and variance (**e**) of subjects' report distributions as a function of inter-target lag. For Guess 2 and 3 panels, trials were split according to Guess 1 position. Closed symbols: subjects reported T-1 as Guess 1; Open symbols: subjects reported T + 1 as Guess 1. Asterisks represent results of a repeated-measures ANOVA with the inter-target lag as a within-subject factor: *$P < .05$; **$P < .01$; ***$P < .001$

studies of subjects' response distributions during RSVP tasks provided important insights on this issue. First, subjects' reports are distributed around the target position, which indicates that subjects often misreport a stimulus either before or after the target[14–16]. Second, the selection depends on the resources devoted to the task as the shape of the report distribution is distorted during dual–task interference[14, 15]. Third, when subjects are asked to produce multiple guesses after each trial, all guesses are distributed around the target position[17], suggesting that subjects can select multiple stimuli on each trial. The goal of the present research is to bring brain-imaging evidence to bear on these mechanisms. We aimed to separate two hypotheses about the brain mechanisms that could explain these behavioral results. First, it could be that attention selects multiple stimuli in parallel, with a gradient of emphasis surrounding the preferred target ("gradual selection"). The alternative hypothesis is that attention selects a single item at a given time, but operates serially over multiple items, enhancing the target stimulus and the surrounding distractors one after the other ("discrete selection", Fig. 1a). Separating those two possibilities is fundamental to understand the overall parallel/serial architecture of vision and visual awareness.

The challenge in separating these models lies in the experimental paradigm itself, where multiple stimuli are presented at a high rate. As a result, brain responses to successive stimuli overlap in time. Here, we address this problem by combining magnetoencephalography with multivariate pattern analyses. In agreement with a previous neurophysiological study in monkeys[18], we find that brain activity, at any instant, contains overlapping but decodable information about several successive images, and we use decoding to separate parallel and serial stages of attentional selection.

Specifically, on each trial, subjects are presented with a rapid stream (stimulus onset asynchrony: 116 ms) of thirteen successive images that can belong to one of five categories (colors, faces, places, body parts, or objects; Fig. 1b). Taking advantage of the well-characterized patterns of brain activity induced by each category, category-selective classifiers are trained to recover the brain responses elicited by each image independently of the other stimuli. Classifiers are trained on data from a localizer task in which the same stimuli are presented one at a time, and are then applied on data from a dual-task experiment in which subjects are instructed to identify two target images (T1 and T2) embedded in the rapid stream (Fig. 1b). By examining how brain activity is amplified for pictures at and surrounding the targets, our goal is to provide a detailed description of how multiple stimuli are simultaneously processed, and how some are selected for further report. The "gradual selection" hypothesis of attentional selection predicts that several images around the target should be simultaneously selected. If this hypothesis is correct, the sensory activity related to multiple stimuli should be simultaneously amplified and sustained by attentional signals. By contrast, the "discrete selection" hypothesis predicts that images are selected one at a time, in correspondence with the subject's report. According to this model, only activations elicited by reported stimuli should be amplified by attention, and their activities should not overlap in time. To anticipate on the results, both patterns were found but at distinct times in the processing pipeline.

## Results

### Distributions of reports in rapid visual streams

Behaviorally, subjects were asked to first determine the color of a patch (T1), and then attempt to identify an image in a rapid visual stream (target T2, surrounded by a green frame). They responded by

making three successive guesses about the target (see Fig. 1b and Method). Decreasing the temporal delay between T1 and T2 limited the attention devoted to T2 and thus allowed us to investigate how selection processes depend on attentional resources. Report accuracy–defined as trials where one of the guesses corresponded to T2–strongly decreased with the inter-target Lag (Repeated-measures analysis of variance (ANOVA), $N = 15$, $F(3,42) = 7.58$, $P < 0.001$), revealing an attentional blink (Supplementary Fig. 1). The full distribution of reports (Fig. 1c) revealed that subjects most often correctly reported the target stimulus (position 'T') as Guess 1 at all lags (Repeated-measures ANOVA, $N = 15$, $F(3,42) = 1281$, $P < 0.001$, Fig. 1d) but also erroneously reported a nearby distractor on a significant number of trials (Fig. 1c). For instance, at lag 9, the stimulus at position 'T-1' was reported more often than chance (One-sample $t$-test, $N = 15$, $t(14) = 2.6$, $P = 0.01$). In fact, the selection processes appeared less precise under attentional constraints, as indicated by an increased variance in Guess 1 at short inter-target lags (Fig. 1e, Repeated-measures ANOVA, $N = 15$, $F(3,42) = 27.89$, $P < 0.001$). Regarding Guess 2, the distribution was centered on stimuli close to and preceding the target at long but not at short lags (Lag 9: Mo = 7, SD = 2.49; Lag 7: Mo = 5, SD = 2.06; Lag3: Mo = 5, SD = 3.42; Lag 1: Mo = 5, SD = 3.45; Repeated-measures ANOVA, $N = 15$, effect of inter-target lag: $F(3,42) = 1.66$, $P = 0.19$). At lag 3, the frequency of report was similar for all positions–over the group, only the target position was marginally above chance (One-sample $t$-test, $N = 15$, $t(14) = 1.77$, $P = 0.049$). Similarly, at lag 1, only the stimulus following the target had a frequency of report above chance (One-sample $t$-test, $N = 15$, $t(14) = 2.75$; $P = 0.008$). The variance of Guess 2 increased with decreasing lag although this effect was not significant (Fig. 1e, Repeated-measures ANOVA, $N = 15$, $F(3,42) = 3.63$, $P = 0.13$). Finally, the distribution of Guess 3 was not as clear cut as for Guess 1 and 2. The only notable pattern was that stimuli at positions T−2 for lags 3, 7, and 9, and at position T + 2 for lag 1 had a frequency of report slightly above chance level (One-sample $t$-test, $N = 15$, all $P < .05$). These results support the idea that more than one stimulus can be selected at each trial[17], and that resource depletion degrades the efficacy of selection processes and makes distractors more likely to interfere.

Although the distribution of reports was on average centered on the target position, it could be that the locus of attention varied from trial to trial and that the successive guesses are centered on this locus. Alternatively, the guesses might reflect samples from a single distribution systematically centered on the target location on each trial. To separate these hypotheses, we examined whether the position of Guess 1 systematically influenced subsequent guesses. When participants erroneously reported the stimulus T + 1 as their Guess 1, the stimulus most often reported as Guess 2 was the target stimulus (Fig. 1d and Supplementary Fig. 2, Repeated-measures ANOVA, $N = 15$, $F(3,36) = 9.8$, $P < 0.001$). A similar but not significant effect was observed when subjects reported the stimulus T-1 as Guess 1 (Repeated-measures ANOVA, $N = 15$, $F(2,28) = 1.71$, $P = 0.2$). This suggests that there was no influence of Guess 1 on Guess 2 (Fig. 1d and Supplementary Fig. 2). As for Guess 3, all stimuli had similar report frequencies, whether subjects reported stimulus T − 1 or T + 1 as Guess 1, except for the stimulus at target position in lag 9 condition when subjects reported stimulus T − 1 as their first guess (Supplementary Fig. 2, One-sample $t$-test, $N = 15$, $t(14) = 2.44$; $P = 0.01$). Those results, together with earlier ones[17], are compatible with the hypothesis that guesses 1 and 2 behave as independent samples from the same distribution. As noted in the introduction, however, they are consistent both with the hypothesis that attentional selection operated gradually and in parallel over multiple stimuli surrounding the target position, and

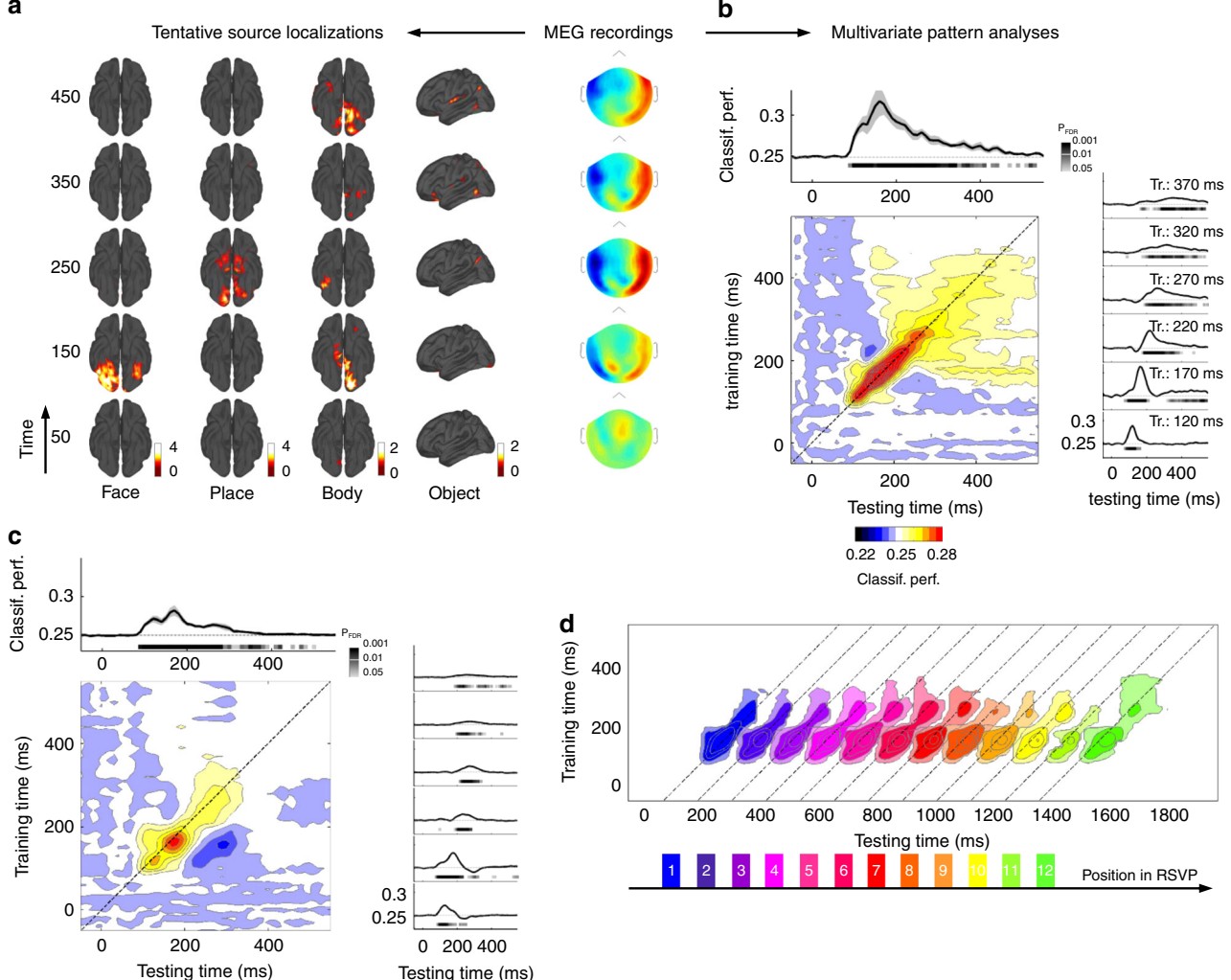

**Fig. 2** Time-resolved decoding of stimulus categories. **a** Right: Example of event-related magnetic fields elicited by the presentation of images during the localizer task. Left: Subtraction between brain sources of each category (face, place, body parts and objects) and the mean of the other categories presented in Z scores according to baseline and projected on a flattened standard brain. Left to right columns represent bottom and left views of the brain at specific time points (50–450 ms from bottom to top). **b** Performance of classifiers trained to separate the four stimulus categories. Top graph: One classifier was trained at each time sample and tested on the same time sample. The shaded gray area represents the standard error to the mean across subjects. Right graphs: Classifiers were trained at specific time samples (from 120 to 370 ms) and tested on all other time samples. Results from signed rank tests comparing classification performance to chance are represented by the thick line below the x axis with darker colors representing lower P-values. False discovery rate (FDR) correction for multiple comparisons was applied across specific training times (120 to 370 ms), all testing times and conditions (RSVP and localizer). Matrix plot: Classifiers were systematically trained on each time sample and tested on all others. The color code represents the classification performance and the dotted line the diagonal of the matrix. **c** Classifiers trained on the localizer task were directly applied to the RSVP task. Line plots, matrix plot and statistics are as in **b**. **d** Contour plot representing classification performance significantly above chance obtained for each stimulus and averaged across subjects. Colors represent the successive stimuli in the RSVP (except T1) depicted below by a small rectangle. Dotted lines represent the diagonals of each matrix plot. FDR correction for multiple comparisons was applied across all training times, testing times and stimulus positions

with the opposite hypothesis that the reported items were recovered serially in a discrete manner.

**Decoding category-selective brain responses.** MEG recordings during the localizer task revealed activity originating in the visual cortex and in the ventral part of the visual stream ~ 100–150 ms after stimulus onset (Supplementary Fig. 3). This activity propagated to the posterior parietal cortex mainly in the right hemisphere. By 450 ms, late activations were found in the orbitofrontal cortex. The systematic comparison of each category against the others revealed category-specific patterns of activity evolving in time, as depicted in Fig. 2a. For instance, at ~ 150 ms, face stimuli induced strong bilateral activations in the fusiform

and occipital cortices while body part stimuli activated the left occipital and inferotemporal cortex.

Multivariate pattern analyses provided further insights on the dynamics of category-specific brain responses. Fig. 2b shows that all classifiers trained between 90 and 530 ms performed above chance level (signed rank tests, $N = 15$, all $P_{FDR} < 0.05$) with a maximum classification performance observed at 160 ms (M = 0.32, SD = 0.7; chance = 0.25). Confusion matrices derived from the decoding analyses revealed that classifiers were specific to a category and no systematic overlap between categories was observed (Supplementary Fig. 4a). Some classifiers performed better than others (e.g., mean classification performance between 150 and 200 ms: 47%, 43%, 32% and 38%, respectively for face, places, objects and body parts, ANOVA on aligned rank

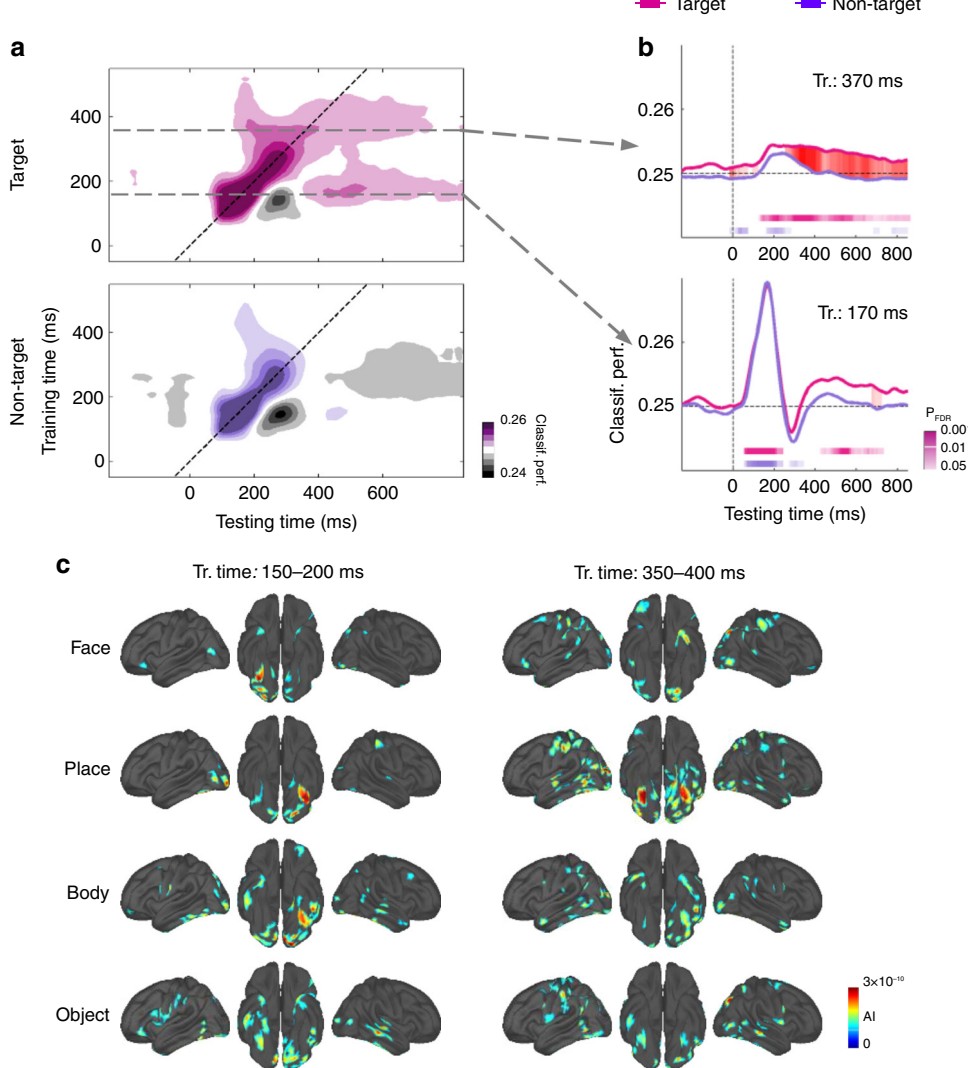

**Fig. 3** MEG evidence for two types of target-specific brain responses: sustained activation of sensory processes, and access to an additional late processing stage. **a** Temporal generalization matrices for target (upper panel) and non-target (lower panel) stimuli averaged across inter-target lags. Gray areas correspond to below chance classification performance. Dashed arrows represent selected training time depicted in **b**. **b** Classification performance as a function of time for classifiers trained at 170 and 370 ms. Results from signed rank tests comparing classification performance to chance are represented by the thick line below the x axis. Signed rank tests comparing target and non-target conditions are represented by areas filled with red, darker colors representing lower P-values. FDR correction for multiple comparisons was applied across specific training times (120–370 ms), testing times and conditions. **c** Source localization of activation patterns computed from the support vector machine (SVM) weights (see method) indicating the location of the most informative activity for each stimulus category; 200–250 ms and 350–400 ms averaged training times are represented. For display purposes, data points were smoothed using a moving average with a window of three samples

transformed data, $N = 15$: F(3,42) = 20.55, $P < 0.001$). However, they all had similar onsets and offsets (defined as the first and last points exceeding the 50th percentile of the distribution, measured on group averaged data): 110–470, 90–450, 120–450, and 90–500 ms, respectively for face, place, object, and body parts categories (Supplementary Fig. 4b).

To unravel the full dynamics of the underlying sequence of processes, each classifier was systematically tested for generalization across time[19] (see Methods section). The ability of a classifier to perform above chance at other time samples allows estimating the latency and the duration of a given pattern of activity. As can be seen in Fig. 2b, classifiers' temporal profiles exhibited a mixture of a diagonal and a square shape. Classifiers at 120 ms had above-chance performance over a relatively short period (80–160 ms, signed rank tests, $N = 15$, all $P_{FDR} < 0.05$). At 170 ms, the time course exhibited a sharp peak between 80 and

210 ms and then decayed down to chance. Interestingly, the same classifier again performed above chance from 320 ms up to the end of the epoch, indicating that the patterns of brain responses had similar features during these two periods of time. This biphasic pattern, suggesting a late reactivation, seemed unique to the classifiers trained between 170 and 210 ms. Beyond ~ 300 ms, classification performance decreased, but the time course remained steadily above chance level up to the end of the trial (signed rank tests, $N = 15$, all $P_{FDR} < 0.05$). Note that classifiers trained at 170 ms and applied at late latencies ( > 400 ms) had better performance than classifiers trained and tested at these late latencies. This suggests that the two classifiers shared similar features but the signal-to-noise ratio (SNR) at 170 ms was better than at late latencies.

The dynamics revealed by the decoding analyses together with the patterns of activity observed at the source level show that

stimulus categories were processed through a series of cognitive operations, starting with fast and short-lived processes in the ventral visual cortex, and ending by stable and sustained activations involving visual and parietal areas. Next, the very same classifiers were directly applied to the RSVP data in order to examine how the sequence of brain processes was affected by the high presentation rate.

**All stimuli undergo the same parallel processing pipeline.** Classifiers trained in the localizer task were applied to RSVP data in order to track the processing stages of each stimulus presented in the RSVP. The results show that stimulus categories can be decoded from 90 ms after stimulus onset up to 520 ms (signed rank tests, $N = 15$, all $P_{FDR} < 0.05$, Fig. 2c). The classification performance was however lower during RSVP than during the localizer task between 100 and 530 ms (all $P_{FDR} < 0.05$, Supplementary Fig. 5). The generalization of classifiers across time revealed dynamics similar to what was observed during the localizer task. Classifiers trained at early latencies ( < 170 ms) had similar onsets and offsets during RSVP and during the localizer. For instance, a classifier trained at 120 ms performed above chance from 90 to 190 ms in the RSVP task and from 80 to 160 ms in the localizer task. At 170 ms, classifiers performance was above chance from 80 to 230 ms as in the localizer task but the second, sustained phase of activity was not observed (Fig. 2c and Supplementary Fig. 5). During this second phase of activity, the classification performance was lower in the RSVP task compared to the localizer task from 270 ms up to the end of the epoch (signed rank tests, $N = 15$, all $P_{FDR} < 0.05$) and barely different from chance level. In fact, the mean classification performance between 400 and 550 ms was lower compared to the localizer task (signed rank test, $N = 15$, $W = 97$; $P = 0.03$) and not different from chance level (signed rank test, $N = 15$, $W = 72$, $P = 0.52$). Finally, brain responses at later latencies had mainly smaller amplitude and shorter durations (Fig. 2c and Supplementary Fig. 5). For instance, classifiers trained at 320 ms performed above chance between 130 and 480 ms during the localizer task but mainly between 150 and 310 ms during the RSVP (signed rank tests, $N = 15$, all $P_{FDR} < 0.05$). The classification performance was significantly stronger in the localizer task compared to the RSVP between 170 and 440 ms (signed rank tests, $N = 15$, all $P_{FDR} < 0.05$). These results show that part of the brain networks solicited during the localizer task were also recruited during RSVP even though the amplitude of brain responses was lower.

Figure 2d shows an overlay of the temporal generalization matrices for each of the 12 successive images following T1 in the rapid stream. A first striking aspect in this figure is that it is possible to decode the brain responses to each stimulus separately from the others. Second, the processing of each stimulus is highly systematic, starting at ~ 90 ms and ending around ~ 350 ms. Third, it appears that at any given moment in a trial, the brain activity contains multiple overlapping codes for distinct stimuli at different stages. For instance, the activity at 1190 ms after RSVP onset already contains decodable information about the 9th image: classifiers between 80 and 200 ms performed above chance. However, the 8th and 7th image can still be decoded from the same data, but only using classifiers trained at later stages (150–250 ms and 270–330 ms respectively, signed rank tests, $N = 15$, all $P_{FDR} < 0.05$). Interestingly, even stimuli presented after the second target (i.e., after the two tasks were completed) were processed just like any other stimulus in the stream, thus demonstrating that these processes are task-independent. These findings suggest the existence of a "pipeline" of neural processes which is automatically deployed each time a stimulus is presented (independently of the task being performed), even if the subject's attention is focused elsewhere. We next examined what brain processes characterized target selection.

**Target stimuli elicit sustained brain activations.** In order to track target-selective brain responses, we measured the classification performance for target stimuli at position 1, 3, 7, and 9 and compared it to trials where the stimulus at the very same position was not a target. Figure 3a shows the classification performance for target and non-target stimuli averaged across lags. The category of the target was decodable from 70 ms until the end of the epoch while the category of non-target stimuli could be decoded only from 60 ms to 340 ms (signed rank tests, $N = 15$, all $P_{FDR} < 0.05$). Classification performance was significantly stronger for target than for non-target stimuli from 330 to 420 ms (signed rank tests, $N = 15$, all $P_{FDR} < 0.05$). This indicates that a stimulus undergo additional processing stages when it is relevant to the task.

Temporal generalization revealed that classifiers trained at 120 ms exhibited a sharp peak of performance that was similar for target and non-target stimuli. At a training time of 170 ms, a biphasic response was observed for target stimuli with a first response between 70 and 230 ms and a second, sustained one between 440 and 720 ms (Fig. 3b, signed rank tests, $N = 15$, all $P_{FDR} < 0.05$). This pattern is similar to what was observed in the localizer task and shows that, when the stimulus was a target, the networks in the two phases of response shared similar features. By contrast, when the stimulus was not a target, only the first phase was observed (signed rank tests, $N = 15$, all $P_{FDR} > 0.11$). The classification performance was steadily better for target compared to non-target stimuli from 530 ms to the end of the epoch (signed rank tests, $N = 15$, all $P_{uncorr} < 0.05$) although this effect barely survived corrections for multiple comparisons (signed rank tests, $N = 15$, $P_{FDR}$ ranging from 0.04 to 0.36). A similar biphasic response was observed for classifiers trained at 220 ms with a first phase between 90 and 310 ms and a second one between 520 and 570 ms (signed rank tests, $N = 15$, all $P_{FDR} < 0.05$), and stronger classification performance for target than for non-target stimuli between 510 and 590 ms (signed rank tests, $N = 15$, all $P_{FDR} < 0.05$). Classifiers trained at 270 and 320 ms revealed similar responses for target and non-target stimuli although target stimuli induced slightly stronger responses between 480 and 500 ms (signed rank tests, $N = 15$, all $P_{FDR} < 0.05$). A marked difference was observed for classifiers trained at 370 ms: for target stimuli, classifiers performed above chance from 140 ms up to the end of the epoch, and significantly stronger than for non-target stimuli from 270 to the end of the epoch. By contrast, non-target stimuli induced only short brain responses from 180 and 260 ms (Fig. 3b, signed rank tests, $N = 15$, all $P_{FDR} < 0.05$). These results suggest that, following initial automatic processing, target images induced additional brain activity that involved both a reactivation of early stages and additional late stages that were sustained over time.

Even though the spatial precision of the MEG signal is limited, we attempted to localize the brain areas that provided target-related informative activity. To this end, the classifiers' weights were projected into an interpretable source activation space (see method). We found that the location of the informative activity depended not only on the category of the presented stimulus, but also on the processing stage (Fig. 3c). At early stages (150–200 ms), the information was located in high-order visual areas such as the right fusiform gyrus for face stimuli or the left fusiform and parahippocampal cortices for place stimuli. During late processing stages (350–400 ms), decodable information was much more scattered. The posterior parietal cortex, the visual cortex and to a lesser extent the frontal cortex contributed to the classification of

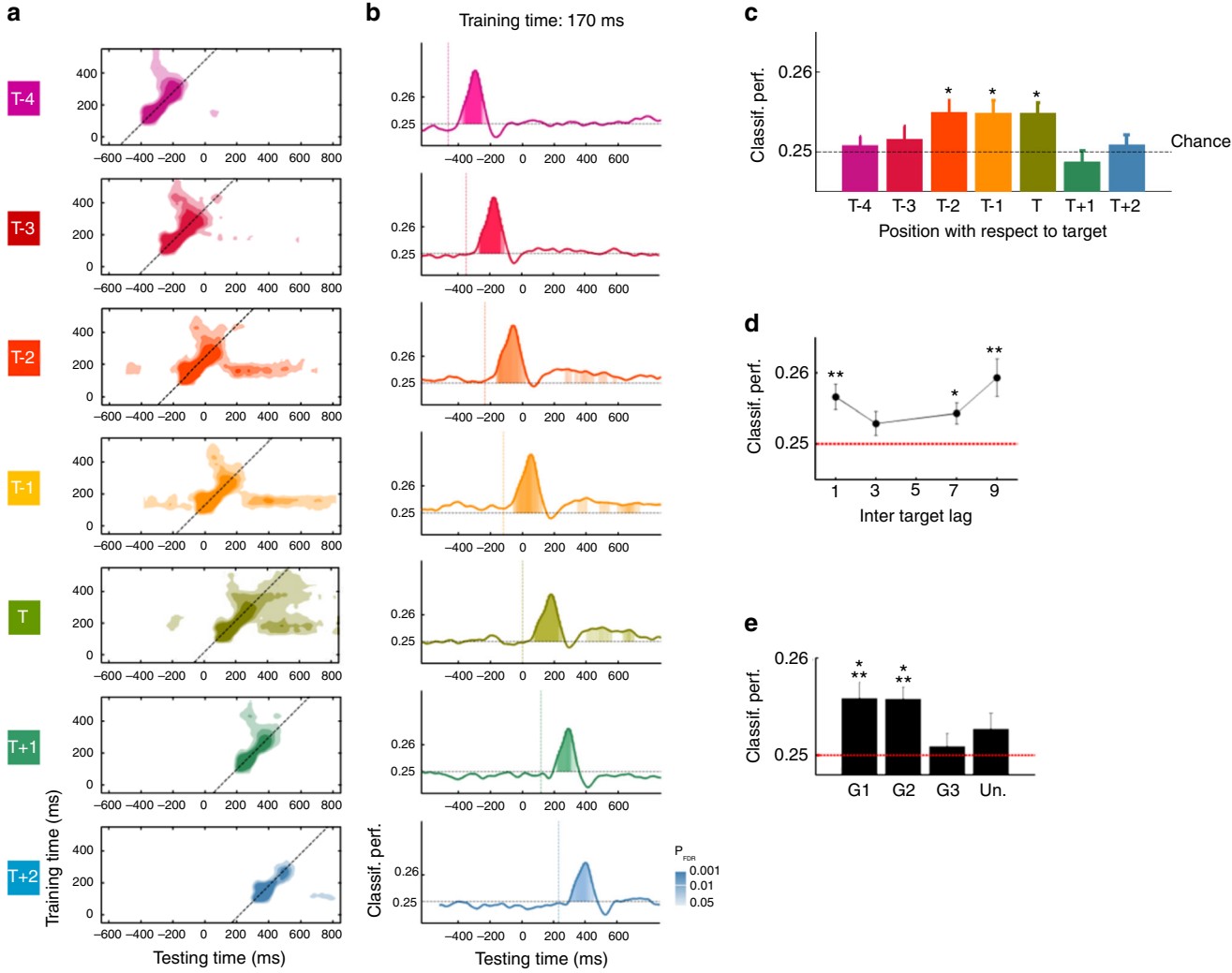

**Fig. 4** Evidence for early gradual selection: early brain responses are selectively sustained over time as a function of target proximity. **a** Panels represent temporal generalization matrices for the target stimulus (T) and distractors at positions T−4 to T + 2 at long inter-target lag. The color scale is as in Fig. 3a except that below-chance classification performance is not represented for visibility purposes. **b** Classification performance as a function of time for a classifier trained at 170 ms after stimulus onset. Filled areas represent time samples at which the classification performance was significantly different from chance (Signed-rank tests, corrected for multiple comparisons using FDR across stimulus positions, training times (170 and 370 ms) and testing times). Colored dotted lines represent stimuli onsets. **c** Mean classification performance of the same classifier but averaged over time (400–550 ms). The dotted line represents chance. **d** Effect of inter-target lag. Classification performance averaged over Guess 1 and 2 and over time (400–550 ms) as a function of inter-target lag. **e** Effect of report. Mean classification performance between 400 and 550 ms for stimuli reported as Guess 1, 2, 3 and unreported stimuli. The red dotted line represents chance. Asterisks represent results from signed rank tests (*$P_{FDR} < 0.05$; **$P_{FDR} < 0.01$; ***$P_{FDR} <0.001$, FDR corrected across stimulus positions). For display purposes, data points were smoothed using a moving average with a window of three samples

stimulus categories. For instance, information regarding face targets were found not only visual areas (occipital and infero-temporal cortices) but also in parietal (inferior and superior parietal cortex) and frontal areas (superior frontal gyrus, central and precentral areas).

These results reveal that target processing involves (i) a pipeline of automatic sensory processes combined with (ii) specific maintenance of early sensory activity, and (iii) late task-related sustained activity. Next, we attempted to better characterize the properties of early and late target-specific activity in order to distinguish the gradual vs discrete selection hypotheses.

**Early target responses reflect gradual selection.** According to the gradual selection hypothesis, multiple stimuli are selected on each trial. This hypothesis predicts that sustained brain responses should be observed for both the target and temporally nearby

distractors. We focused our analyses on stimuli at positions T−4 to T + 2, where T is the ordinal position of the target, averaged across lags 7 and 9.

Classifiers trained at 170 ms revealed biphasic responses for stimuli at positions T−1 and T−2, i.e., a first sharp peak followed by a sustained activity extending over several hundred milli-seconds, which were highly similar to the one observed for target stimuli (Fig. 4a). Specifically, relative to target onset, the stimulus at position T−2 elicited above-chance classification performance between −152 and + 8 ms, and between 278 and 578 ms. Similarly for the stimulus at position T−1, we observed above-chance classification performance from −46 to 114 ms and from 354 to 724 ms. Finally, above-chance classification performance for the target stimulus was observed from 80 to 240 ms and from 410 to 690 ms (signed rank tests, $N = 15$, all $P_{FDR} <0.05$, see Fig. 4b). Thus, both the peak and the sustained phase of activity followed the order of presentation of the stimuli. Crucially however, the

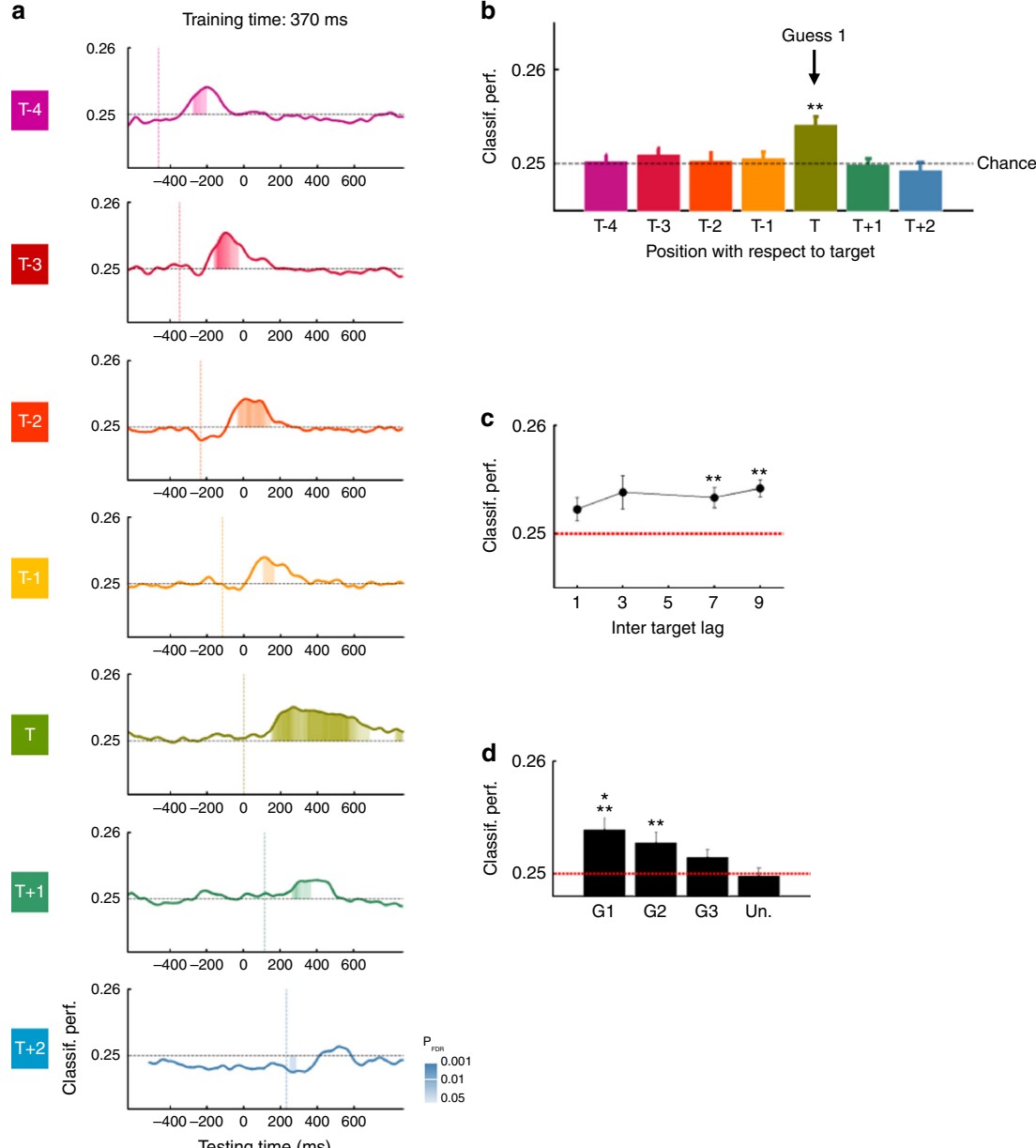

**Fig. 5** Evidence for late discrete selection: Late brain responses are sustained only for reported stimuli. **a** Classification performance as a function of time for target stimulus and nearby distractors for a classifier trained at 370 ms (colors and statistics are as in Fig. 4b). Colored dotted lines represent stimuli onsets. **b** Mean classification accuracy over a time period of 400–550 ms (as in Fig. 4c). **c** Effect of inter-target lag. Classification performance averaged over Guess 1 and 2 and over time (400–550 ms) as a function of inter-target lag. **d** Effect of report. Mean classification performance between 400 and 550 ms for stimuli reported as Guess 1, 2, 3, and unreported stimuli. The red dotted line represents chance. Asterisks represent results from signed rank tests (*$P < 0.05$; **$P < 0.01$; ***$P < 0.001$, FDR corrected across conditions). For display purposes, data points were smoothed using a moving average with a window of three samples

second phase of activity occurred after target onset for all three stimuli. Furthermore, at least between 410 and 578 ms (relative to target onset), category information for stimuli T−2, T−1 and T could be decoded simultaneously and from the same data with classifiers trained at 170 ms. This is evidence for gradual, parallel processing in the early stages of temporal selection.

The average classification performance between 400 and 550 ms after stimulus onset was above chance for stimuli at positions T, T−1, and T−2 (Fig. 4c, signed rank tests, $N = 15$, $W = 108$, $P_{FDR} = 0.028$; $W = 103$, $P_{FDR} = 0.04$ and $W = 101$, $P_{FDR} = 0.047$ respectively). Distractors at positions T−1 and T−2 induced responses similar to the one observed for target stimuli

(T vs T−2, signed rank tests, $N = 15$: $W = 58$, $P_{FDR} = 0.93$ and T vs T−1: $W = 62$, $P_{FDR} = 0.93$, respectively). For each of these stimuli, the classification performance seemed slightly stronger compared to stimuli at positions T−4 (signed rank tests, $N = 15$, all $P_{FDR} < 0.056$), T + 1 (signed rank tests, $N = 15$, all $P_{FDR} < 0.05$), and T + 2 (signed rank tests, $N = 15$, all $P_{FDR} < 0.1$). To further test that this effect reflects within-trial modulations and not between-trial variance, we conducted the same analyses only on trials where subjects correctly reported the target stimulus as Guess 1. Although the reduction in the number of trials limits the statistical power of this analysis, we found that the average classification performance between 400 and 550 ms after stimulus

onset was significantly above chance for stimuli at positions T and T-2 when considering uncorrected $p$-values (signed rank tests, $N = 15$, $W = 95$, $P_{uncorr} = 0.048$ and $W = 99$, $P_{uncorr} = 0.026$). Stimuli at positions T, T$-1$, T$-2$ also elicited stronger responses compared to stimuli at position T$+1$ (signed rank tests, $N = 15$, all $P_{uncorr} < 0.05$). No difference was found with stimuli at position $-4$ (signed rank tests, $N = 15$, all $P_{FDR} > 0.15$) and $+2$ (signed rank tests, $N = 15$, all $P_{FDR} > 0.11$). Together, these results suggest that target-selection processes affected simultaneously the processing of multiple nearby distractors. This is consistent with the gradual selection hypothesis in which attention would affect the processing of multiple stimuli simultaneously on each trial.

To test whether the observed sustained activations were related to subjects' attention and reports, we examined the effects of inter-target lag and subjects' order of guesses on the mean classification performance between 400 ms and 550 ms ms for each guess and for unreported stimuli (i.e., a stimulus not reported by subjects that was randomly selected on each trial). We found a main effect of inter-target lag (Repeated-measures ANOVA on aligned rank-transformed data, $N = 15$, main effect: $F(3,42) = 4.38$, $P = .009$). When averaged across Guess 1 and Guess 2, which corresponded to actual report of the subject rather than random guessing, the classification performance varied as a U-shape function of inter-target lag (Fig. 4d). The classification performance decreased from lag 9 to lag 3 (signed rank tests, $N = 15$, $W = 17$, $P_{FDR} = 0.036$). In fact, the performance at lag 3 was not different from chance (signed rank tests, $N = 15$, $W = 84$, $P_{FDR} = 0.72$), unlike the performance in the other lag conditions (signed rank tests, $N = 15$, $W = 114$, $P_{FDR} = 0.008$, $W = 105$, $P_{FDR} = 0.033$ and $W = 112$, $P_{FDR} = 0.008$ for lag 1, 7, and 9, respectively). This result shows that these sustained brain responses were affected by attentional constraints and supports the idea that these are linked to attentional modulations.

Next, we examined the link between sustained brain responses and subjects' behavior. The main effect of guesses did not reach significance (Fig. 4e, Repeated-measures ANOVA on aligned rank-transformed data, $N = 15$, $F(3,42) = 2.33$, $P = 0.09$) and there was no significant interaction with the inter-target lag (F(9126) = 1.64, $P = 0.11$). However, the classification performance was significantly above chance for Guess 1 and 2 (signed rank tests, $N = 15$, $W = 112$, $P_{FDR} = 0.01$ and $W = 117$, $P_{FDR} = 0.006$, respectively) but not for Guess 3 and unreported stimuli ($W = 67$, $P_{FDR} = 0.72$ and $W = 84$, $P_{FDR} = 0.31$, see also Supplementary Fig. 5). In fact, Guess 1 and 2 induced responses of similar amplitudes ($W = 68$; $P_{FDR} = 0.72$), slightly stronger than Guess 3 (G1 vs G3: $W = 96$, $P_{FDR} = 0.1$; G2 vs G3: $W = 103$, $P_{FDR} = 0.04$). Finally, Guess 3 did not differ from unreported stimuli ($W = 39$, $P_{FDR} = 0.38$). These results show that these sustained brain responses were only partially related to subjects' behavior: although they were related to the selected stimuli, they did not reflect subjects' order of preference.

Together, these results reveal that temporal selection first proceeds through a gradual attentional enhancement centered on target position, but spreading to temporally nearby stimuli. Temporal attention can thus affect the processing of multiple stimuli in parallel.

**Late target responses reflect discrete selection.** Target stimuli did not only elicit a maintenance of initial brain responses but also a late activity ($> 350$ ms) originating from visual, parietal and frontal areas. However, this activity was not observed for nearby distractors (Figs. 4a and 5a). In fact, at 370 ms, the brain responses elicited by these stimuli were only transient and short-lived. Only the reported target stimulus elicited a strong and sustained activity (Fig. 5a, signed rank tests, $N = 15$, all

$P_{FDR} < 0.05$). Examining the average classification performance between 400 and 550 ms after stimulus onset revealed that the response to the target stimulus was stronger than the ones observed for all other stimuli (signed rank tests, $N = 15$, all $P_{FDR} < 0.02$) and the only one above chance (signed rank tests, $N = 15$, $W = 120$, $P_{FDR} = 0.001$, Fig. 5b). This is in contrast to what was observed with classifiers trained at 170 ms for which we observed an effect of the temporal proximity of the target stimulus.

To investigate whether these late brain responses were influenced by attentional constraints, we tested the effects of inter-target lag on the mean classification performance over a period of 400–550 ms after stimulus presentation for each guess and for unreported stimuli. We found a significant main effect of inter-target lag (Fig. 5c, Repeated-measures ANOVA on aligned rank-transformed data, $N = 15$, $F(3,42) = 7.89$, $P < 0.001$). When averaged across Guess 1 and 2, the performance was significantly lower in lag 1 condition compared to lag 9 (signed rank tests, $N = 15$, $W = 14$, $P_{FDR} = 0.027$). No other difference between lag conditions was observed (all $P_{FDR} > 0.11$). The performance at lag 1 and 3 did not significantly differ from chance ($W = 91$, $P_{FDR} = 0.14$; $W = 96$, $P_{FDR} = 0.1$), while it was above chance for long lag conditions ($W = 114$, $P_{FDR} = 0.006$; $W$ $W = 120$, $P_{FDR} = 0.001$ for lag 7 and 9, respectively). Thus, attentional constraints also affected late stages of temporal selection.

We next examined the effect of subjects' behavior. We found a main effect of Guesses on classification performance (Repeated-measures ANOVA on aligned rank-transformed data, $N = 15$, $F(3,42) = 3.23$, $P = 0.03$) but no interaction with the inter-target lag (F(9126) = 1.34, $P = 0.22$). When averaged across lag conditions, the classification performance was significantly stronger for Guess 1 than unreported stimuli (signed rank tests, $N = 15$, $W = 111$, $P_{FDR} = 0.01$). Similar but weaker differences were observed for Guess 2 and Guess 3 compared to unreported stimuli (Guess 2: $W = 97$, $P_{FDR} = 0.1$; Guess 3: $W = 95$, $P_{FDR} = 0.1$). In fact, the performance linearly decreased from Guess 1 to unreported stimuli (Fig. 5d, mean slope of a linear regression with Guesses: M $= 1.4e^{-3}$, s.e.m.: $4.64e^{-4}$; signed rank tests, $N = 15$, $W = 110$, $P = 0.003$). Only Guess 1 and 2 elicited reliable sustained activity (Fig. 5d and Supplementary Fig. 6, Guess 1: $W = 119$, $P_{FDR} = 0.001$; Guess 2: $W = 105$, $P_{FDR} = 0.028$; Guess 3: $W = 93$, $P_{FDR} = 0.11$; Unreported: $W = 46$, $P_{FDR} = 0.48$). This shows that the multiple guesses produced by subjects were differentiated mainly by the amplitude of late brain responses in a distributed network including visual, parietal and, to a lesser extent, frontal areas.

Taken together, these results show that, among the stimuli that benefit from attentional enhancement, only those that subjects eventually report undergo an additional late processing stage. Both gradual and discrete selection exist: target selection processes starts with a gradual amplification of early brain responses and ends with late discrete stages tightly linked to subjects' order of preference.

## Discussion

The present study aimed at understanding how the brain processes, selects and gates relevant information to awareness when it is bombarded by visual inputs. Previous studies using RSVP have shown that the efficiency of brain areas to process external information varies along the ventral stream, with lower-order areas processing information at a faster rate than higher-order areas[20, 21]. In the present study, we explored the dynamics of the selection mechanisms. Specifically, we tested whether temporal selection operates on multiple stimuli in parallel ("gradual selection") or whether stimuli are selected one after the other ("discrete selection"). So far, the existence of such mechanisms in the brain was only indirectly suggested[15, 17]. The present study provides

strong empirical evidence that both hypotheses are true but occur at different times. When multiple stimuli are flashed, the brain is able to efficiently process each stimulus. The multiple representations coexist in the visual cortex for several 100 ms, but at different processing levels within the same pipeline. When one of these representations is task-relevant, the selection starts by a continuous attentional enhancement of multiple stimuli around the target position. Then, a subset of these stimuli accesses additional discrete processes that allow conscious report. These results suggest that temporal selection involves at least two successive operations: a parallel probabilistic selection followed by a serial sampling. This provides new perspective on how the human brain reliably extract relevant information from overwhelming external inputs.

The high spatial selectivity of decoding algorithms allowed us to separate, for the first time to our knowledge, category-related brain responses to each stimulus in a rapid visual stream. The results revealed that all stimuli elicited a sequence of brain responses with highly similar temporal profiles. This suggests that these stages of processing can be deployed in parallel to another task, even under attentional constraints. It does not, however, imply that each of these early stages can process multiple stimuli simultaneously. Rather, our data show that the representations of multiple stimuli can coexist in sensory areas but at different stages of processing. Altogether, these results suggest that stimuli presented at high presentation rate are systematically integrated via an automatic, parallel and effortless pipeline of processes.

Our results revealed that the classification performance was overall smaller in the RSVP task than in the localizer task, probably due to a partial masking effect[22]. Importantly, we also found that the sustained activations in visual areas observed in the localizer task were specifically disrupted during RSVP, at least for non-target stimuli. Previous studies have shown that backward masking leaves intact the early feedforward activity of V1 neurons and mainly disrupts the later and sustained part of the activity linked to feedback attentional signals from higher areas[23–25]. This is consistent with monkey studies which showed that the effects of attention on higher-order visual areas are essentially observed between 100 and 300 ms after stimulus onset[26]. The present results suggest that, for irrelevant stimuli, feedback attentional signals were disrupted during RSVP, while feedforward processing was left unaffected. To what extent sensory information continues to be processed normally through these feedforward processes during RSVP remains an open question. Our data merely indicates that it is processed up to the categorical level ("gist recognition"), but this need not imply that precise identification was achieved. There is evidence that feedforward processing allows the brain to quickly extract complex information such as category[6] or meaning[5] from rapidly changing visual inputs. By contrast, as further discussed below, only the stimuli surrounding the target elicited additional processes including feedback attentional signals and were processed up to complete identification.

When a target stimulus was presented, a specific activity in the visual cortex was elicited around 150–200 ms and sustained over several 100 ms. This activity was modulated by the amount of attention devoted to Task 2, and observed simultaneously for the target stimulus and for nearby distractors during an extended period of time. This rejects the hypothesis of a discrete selection mechanism and rather shows that temporal attention gradually selects multiple stimuli in parallel on each trial. That stage may therefore correspond to the forming of a continuous probability distribution over potential target stimuli, from which subsequent reports are sampled, as inferred from earlier behavioral experiments[17]. The width of this distribution, and thus the number of stimuli affected by attention, might not be fixed and could instead depend on the task. It is plausible for instance that the attentional time window would shrink or expands depending on subjects' goal.

In contrast with early selection processes, late brain responses (370 ms) exhibited a discrete profile. Sustained brain responses in visual, parietal and frontal areas were specifically observed for the target stimulus while nearby distractors elicited only short-lived and transient activity. This sustained activity was only observed for reported stimuli, and its amplitude decreased under attentional constraints. Furthermore, this activity also reflected subjects' order of guesses. This is fully consistent with the idea of a discrete, all-or-none sampling process and rejects a gradual selection. Assuming that subjects only reported stimuli that they were conscious of, these late "discrete" brain responses should be related to either the conscious representation of the stimulus or its gating into awareness. Taken together, the present results show that the ability of the brain to select relevant information among multiple coexisting representations relies on two successive operations: first, a global and non-specific attentional enhancement of the multiple representations present in the sensory cortex; second, a discrete maintenance of a subset of these representations, consciously reportable, and weighted by their probability of corresponding to the target stimulus.

Our interpretation of the present findings builds on previous research which provided evidence that representations of multiple stimuli are actively maintained in a perceptual buffer and compete for access to awareness[10, 11, 27–29]. Eventually, one of these representations will trigger the activation of all-or-none conscious processes[30–32]. When resources are limited (i.e., when the inter-target lag is short), the attentional amplification is weaker and more distractors are likely to win the competition. In that case, the identification of the target stimulus might be degraded or alternatively, a distractor stimulus could be erroneously bound to the target feature (the green square) and identified in place of the actual target stimulus.

The fact that the number of conscious reports that subjects can produce in a single trial is limited (around 2 or 3)[17] might result from the serial characteristic of conscious access[10, 11]. The conscious representation of a stimulus takes time. Because the sensory information in the buffer decays over time, each iteration (i.e. each conscious report) decreases the chance that an additional stimulus will be successfully retrieved.

The present data also has implications for current theories of dual-task interference. Research on the AB revealed that the visibility of the second target is degraded when the inter-target lag decreases. The AB typically reaches its maximum at lags 2–3 (i.e., 200–300 ms after the first target) but surprisingly subjects' performance recovers at the shortest lag (the so-called 'lag-1 sparing'). It has been debated whether at lag-1 the first and the second target are perceived serially as two separate events, or in parallel as a single perceptual event, which would imply that multiple stimuli can be consciously perceived simultaneously[33–35]. The "boost and bounce" model of the AB for instance proposes that the target stimulus triggers an attentional enhancement which affects not only the target but also the successive stimuli presented at the same location[36]. After this initial "boost" which gates sensory information into working memory, inhibitory feedback signals would "bounce" subsequent irrelevant sensory information (i.e., distractor stimuli)[36]. According to the model, the consolidation of T2 in working memory would only proceed once the attentional set switches from T1 to T2. Therefore, T2 would benefit from attentional enhancement if it shares features with T1, but its consolidation would nevertheless be delayed. According to the model, the attention devoted to T2 at lag-1 is sufficient to explain why its perception is spared. Interestingly, another dominant model of the AB the "simultaneous type, serial

token" (ST$^2$) model[37] makes a different prediction: according to this model, after an initial attentional enhancement, T1 and T2 stimuli would enter awareness simultaneously if T2 is presented at lag−1. T2 perception would therefore be spared at lag−1, not only because it receives attentional signals but also because T1 and T2 access the same conscious episode. The present results are compatible with the Boost and Bounce model and support the serial view: at lag 1, consecutive stimuli benefit from parallel attentional enhancement, but their access to awareness is serial (as also suggested by other studies[11, 29]).

The present findings show how a computational architecture that involves a succession of gradual and all-or-none processes allows the brain to select relevant information even when it is bombarded by visual inputs. We provide evidence that the visual cortex acts as a perceptual buffer in which multiple representations temporarily coexist. Attention allows a gradual maintenance and amplification of a subset of these representations. Late discrete processes samples information from the buffer and broadcast it to consciousness and working memory in an all-or-none manner. Although the present study focuses on the temporal domain, it also connects to other aspects of human cognition. There is evidence that a similar succession of parallel probabilistic selection followed by serial sampling exist in spatial attention tasks[17, 38], sentence comprehension[28], or the chaining of mental operations[39]. Thus, the mechanisms depicted in our study seem to reflect general mechanisms by which the brain selects information and gates it to awareness so that it can be further manipulated.

## Methods

**Subjects**. Sixteen subjects (11 male, 20–35 years of age) with no history of neurological or psychiatric disorders participated to the experiment and received 40 € for their participation. The study was approved by the ethical committee ("Comité de Protection des Personnes") and all subjects gave informed and written consents before the experiment. Subjects were naive with respect to the task and all had normal or corrected to normal visual acuity. One subject did not comply with the task instructions and was therefore excluded from the analyses.

**Stimuli and apparatus**. Images of five categories (colored squares, grayscale images of faces, places, body parts and objects), downloaded from the internet, were presented at the center of the screen on a black background (for copyright reasons, the stimuli used in the experiment are replaced by representative images in Fig. 1b). Stimuli were back projected on a screen (refresh rate: 60 Hz) placed one meter in front of the subject under standard overhead fluorescence lighting. The experiment was controlled by a Pentium IV PC running PsychToolbox 3.0.9 with Matlab 7.11. Subjects' hand responses were recorded with a five button non-magnetic response box (Cambridge Research System Ltd. Fibre Optic Response Pad). Vocal responses were recorded with a microphone placed next to the head of the subjects. The luminance was kept similar between stimuli (M = 29.87 lx, SD = 1.61) to ensure that subjects' attention was not captured by the unusual luminance of one of the stimuli. We also examined whether report accuracy was comparable between categories. No significant effect of Category was observed on report accuracy (Repeated-measures ANOVA, $N = 15$, $F(3,42) = 0.37$, $P = 0.77$).

**Localizer task**. The experiment started with two blocks (150 trials each) of a single-task condition in which only one stimulus was presented on each trial (see Fig. 1a). A trial started with the presentation of a fixation cross (duration: 800–1200 ms) immediately followed by the stimulus (duration: 84 ms, size: width: 12°, height: 7°). The fixation cross then reappeared for 1000 ms after which the response screen was presented. A list of the possible categories was displayed on the screen, each one associated with a specific button (e.g., 1: Face; 2: Place; 3: Body part; 4: Object; 5: Color). Subjects were instructed to report the category of the stimulus by pressing one of the five response buttons with their dominant hand. The relation between response buttons and stimuli categories was randomly shuffled on each trial so that subjects could not anticipate their motor response before the presentation of the response screen.

**Dual-task**. Subjects then performed five blocks (60 trials each) of a dual-task experiment in which, on each trial, 13 stimuli were presented in a rapid visual stream (presentation rate: 8.6 Hz; stimuli onset asynchrony, SOA: 116 ms, stimulus duration: 84 ms, see Fig. 1b). The first task was to identify the color of the very first stimulus of the stream (T1, either a blue or a red square). For each of the following

12 stimuli, one image was randomly selected among 40 possibilities and could either be a face, a place, body part or an object (10 possible stimuli per category). The second task was to identify the stimulus that was surrounded by a green square (T2). T2 was presented at one of four possible positions (1, 3, 7, or 9 with respect to T1, thereafter referred to as Lag 1, 3, 7, and 9).

A trial started with the presentation of a fixation cross (duration: 600–800 ms) followed by the first stimulus (T1) of the RSVP. At the end of the RSVP, the fixation reappeared for 1000 ms. The response screen for Task 1 was then presented. Subjects reported T1 color with their dominant hand by pressing one of two buttons (e.g., 1: red; 2: blue). The association between colors and response buttons was shuffled on each trial. Once a response for Task 1 was recorded, the response screen for Task 2 appeared. All stimuli presented in the RSVP were displayed simultaneously and evenly spaced on the screen. Each image was topped by a letter that was orally named by subjects to indicate their choice. On each trial, subjects made systematically three guesses to identify T2, that is, they orally named three letters, each letter corresponding to one stimulus. Guesses were given by order of preference, the first response corresponding to subjects' first choice for T2 identity. A maximum of five seconds was allowed for Task 2 responses. Again, the positions of the images on the screen were randomly shuffled on each trial so that subjects could not anticipate their response before the presentation of the response screen.

**MEG recordings and preprocessing**. Brain signals were continuously recorded with a 306-channel whole-head magnetometer (Elekta Neuromag®, Sampling rate: 1000 Hz; High pass filter: 0.1 Hz; Low pass filter: 330 Hz) within a room shielded against electromagnetic noise (Maxshield). MEG channels were organized in 102 triplets composed of one magnetometer and two orthogonal planar gradiometers. In addition, electrocardiogram, and vertical and horizontal electro-oculograms were also recorded for offline rejection of artefacts induced by eye movements and heartbeat. Subjects' head positions were tracked with four coils placed over frontal and mastoïdian skull areas and measured at the beginning of each run with an isotrack polhemus Inc. system. Head positions were then realigned on the position of the first run in order to compensate for head movements between runs. Signal Space Separation[40] was applied to MEG signals in order to decrease the impact of magnetic sources outside the sensor helmet. Magnetometers and gradiometers were visually inspected to identify bad channels (1–7 bad channels across subjects). Head movement compensation, bad channel correction and signal space separation were applied using MaxFilter software (Elekta®).

Continuous data were then epoched with the Fieldtrip software[41] (http://www.fieldtriptoolbox.org/). Localizer epochs range from −200 to 600 ms after stimulus presentation. Dual-task epochs started 500 ms before T1 onset and ended 2000 ms later during. A baseline correction was applied for each trial and each sensor using the time period before stimulus onset. A panel of measures (variance, minimum, maximum, range) were then computed across sensors and displayed in scatter plots in order to identify and reject the trials that might be artifacted (mean proportion of rejected trials per subject: M = 4.68%, SD = 3.77). Independent component analyses were applied separately for each type of sensor using fastICA algorithm[42]. Components topographies were visually inspected and their time courses were correlated with the EOG and ECG signals. The components related to the cardiac artefact or to the eye movements were then rejected from the raw data.

**Source localizations**. An anatomical MRI (3T Siemens MRI scanner with a spatial resolution of $1 \times 1 \times 1.1$ mm$^3$) was acquired for each subject. Subjects' head were digitized and their position inside the sensor helmet was tracked in order to co-egister the MEG signals and subjects' anatomy. Gray and white matters were segmented with the Freesurfer software[43, 44] (http://surfer.nmr.mgh.harvard.edu/). Cortical surfaces were reconstructed with Brainstorm©[45]. Models of the cortex and of the head were used to estimate the current-source density over the cortical surface. The forward model was computed with overlapping spheres analytical model. Weighted minimum norm estimate (wMNE) was used for inverse modeling (depth-weighting factor: 0.5) and dipole orientations were constrained to be normal to the cortical mantle. In order to perform group analyses, the source estimate data of each individual were projected on the freesurfer standard anatomical template (an average brain based on 40 subjects). Single subject MEG signals were transformed in Z-scores relative to baseline and spatially smoothed over 10 mm.

MEG, MRI data, and analyses code are available upon request.

**Multivariate pattern analysis**. In order to facilitate data handling and decrease the computation time of decoding analyses, MEG signals were filtered below 30 Hz, down-sampled to 100 Hz, and the epochs used as training data set (the localizer task) was restrained to −50 to 550 ms after stimulus onset. Multivariate pattern analysis were applied in the sensors space at each time sample using the Scikit-learn python package[46]. A 5-fold stratified cross-validation procedure was used for within-subjects analyses. For a given time sample, the MEG data were randomly split into 5 folds of trials and normalized (Z-score of each channel-time feature within the cross-validation). The same proportion of each class was kept within each fold (stratification). A linear support vector machine[47] (SVM) (penalty parameter C fixed to 1) was trained on 4 folds and tested on the left out trials in order to find the hyperplane that best separated the classes without overfitting. A

weighting procedure was applied in order to equalize the contribution of each class to the definition of the hyperplane. The procedure was iteratively applied for each time sample of each fold. The multiclass classification problem was decomposed into multiple binary classification problems using a "one-vs-the-rest" strategy: for each class, one classifier was fitted against all the other classes.

Classifiers trained at each time sample were also tested on their ability to discriminate categories at all other time samples. The complete "temporal generalization"[19] results in a matrix of training time×testing time. The diagonal of this matrix corresponds to classifiers trained and tested on the same time sample. Training one classifier at time $t$ and generalizing it at time $t'$ was performed within the cross-validation so that $t$ and $t'$ data came from independent sets of trials.

Classifiers were trained in the Localizer task to separate four stimulus categories that could correspond to T2 in the Dual-task condition (i.e., face, place, body parts, and objects). These classifiers were then applied to each trial of the dual-task condition.

The weights assigned by classifiers to MEG sensors correspond to the degree to which the signal measured by a given sensor is used by a classifier to separate classes. Interpreting the weights is difficult because a high weight can be assigned either because the sensor provides class-specific information or because it is useful to decrease the noise. In order to project the classifiers' weights into an interpretable activation space, the SVM weights were multiplied by the covariance of the data. In that space, MEG sensors with large amplitudes indicate a high degree of class-specific information (for a full description and discussion of this method, see ref. [48]). Of particular interest, it is possible to apply on these activation patterns standard source-localization methods to identify brain areas that contributed the most to the separation of the classes ("Informative activity") (see Fig. 3c).

**Statistical analyses**. Classification was complemented with a continuous prob-abilistic output[49] representing the probability that the stimulus presented belonged to one of the four T2 categories (the chance level was therefore 0.25). The classification procedure was repeated for each time point of each trial, resulting in a matrix training time×testing time×trials×classes. The probabilities of correct classification were averaged for each stimulus in the RSVP and for each trial, thus resulting in a matrix train time×test time×stimulus. Trials were then averaged according to the condition or report of interest (e.g., Lag 9 trials).

Statistical analyses were performed across subjects using signed rank tests with a threshold of significance set at $\alpha = 0.05$ to assess whether classifiers could predict the trials' classes above the chance level. Comparisons between experimental conditions were performed using repeated-measures ANOVA on aligned rank transformed data (ARTool library for R-software[50]) and signed rank tests. Analyses were performed between 0 (i.e. stimulus onset) and 550 ms for the localizer task and between 0 and 900 ms for each stimulus of the RSVP. Unless otherwise specified, a correction for multiple comparisons was applied (FDR) across testing times, specific training times (120, 170, 220, 270, 320, and 370 ms), conditions (e.g., target and non-target stimuli), and/or stimulus positions. Exact adjusted $P$-values are reported unless it was below 0.001.

**Data availability**. The data that supporting the findings of this study are available from the corresponding author upon reasonable request.

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

## Acknowledgements

This research was supported by the Commissariat à l'Energie Atomique et aux Energies Alternatives (CEA), the Institut National de la Santé et de la Recherche Médicale (INSERM) and a senior grant of the European Research Council—the NeuroConsc program—to S.D. The Neurospin MEG facility was sponsored by grants from INSERM, CEA, Fondation pour la Recherche Médicale (FRM), the Bettencourt-Schueller Foundation, and the Région Ile-de-France. We are grateful to Florent Meyniel, Clément Moutard, Aaron Schurger, and Virginie Van Wassenhove for constructive discussions and to UNIACT team for the recruitment and preparation of subjects.

## Author contributions

Conceived and designed the experiment: S.M., S.D. Performed the experiment: S.M. Analyzed the data: S.M. Contributed analysis tools: S.M. Contributed to the writing of the manuscript: S.M., S.D.

## Additional information

**Competing interests:** The authors declare no competing financial interests.

