## [Peer Review File · Nature Communications]

Reviewers' comments:

Reviewer #2 (Remarks to the Author):

Overall this was a nice, clever study that takes advantage of category decoding methods with MEG to study processing of target and non-target items in an RSVP sequence.

While I like the study very much, I do not think that the data support the strong conclusions about parallel (gradual selection) versus discrete selection. The results do support findings from the senior author's group and other studies showing enhanced processing of target items, in particular after around 300 ms. However, I do not think that these results allow us to discard an interpretation in terms of serial (rather than parallel) mechanism. Overall, I had three main issues along with some other more minor comments.

My first main concern derives from the choice to present the separate the items by about 148 ms in the RSVP sequence (approximately 6.7 Hz). This is long enough for the visual system to complete an initial processing of each image in a serial fashion (as the authors discuss in detail in the Discussion section). Numerous studies have provided evidence that it takes about 100-150 ms for a stimulus to get through the "first pass" of visual processing, with recurrent processing becoming increasingly important around 150-200 ms (in particular, work from people like Breitmeyer, Di Lollo or Wutz with integration masking; work on rapid identification by Thorpe and colleagues).

An alternative explanation for the same data, if it were coming, for example, from Rufin VanRullen's lab, is that the results reflect a 7 Hz attentional sampling or "perceptual cycle" in visual processing. Note that VanRullen's conception of sampling is actually serial, with each cycle occurring sequentially.

Second, I think that the results are more confirmatory than novel. The finding that M/EEG or fMRI decoding of category can be found in the time period of around 100-300 ms and follows behavioral results has been previously shown, as the authors do a good job of pointing out in the paper. In fact, above-chance decoding also even been found for unreported ("invisible") stimuli in other paradigms (orientation or flicker information, etc...). Thus, while it is nice to see that category decoding works at 6.7 Hz, it is not particularly surprising.

There are a number of relevant studies showing evidence for a minimum temporal duration that supports processing of a sequence of images. One nice example is Gauthier et al., J Neurosci, 2012. They varied the RSVP rate (for alternations between two images) between 33 – 4800 ms in an fMRI study. They reported that earlier visual areas responded to fast presentation rates (stimuli up to 100 – 200 ms) while later areas required longer display durations (see also McKeef et al., 2007, showing faster rates in early visual cortex and a limit of around 100-200 ms for FFA and PPA).

I was also worried that the later stage (>300 ms) does likely involve the selection of a target for response. The participant was actually supposed to respond based on the category of the target image. The design did not dissociate the decoded feature from the response, as one might have hoped. Thus, it is certainly possible that at least some aspect of the decoding performance at the later time period was about categorical response selection (linked to parietal and frontal areas) rather than about attentional selection in visual areas.

My third concern is the more general theoretical implications. The authors do not really motivate why performance in an RSVP task is so important and I found the discussion of RSVP to lack precision. In

terms of theory, in the real world, natural stimuli are not very often presented at an RSVP rate of 6.7 Hz, but rather much faster (video) or slower (saccades at 3-5 Hz). I understand that this is a short format, but it is left open what is actually processed (gist recognition) at this rate. On the one hand, it is claimed that it is efficient at 10 Hz, but then claimed that the brain cannot fully process them (whatever that means), and some images are subjectively invisible. But the studies cited cover a wide variety of stimuli, including both words and photographs, and differing amount of masking. Holcombe (2009), for example, has argued for a cut-off at around 10 Hz.

I would be worried that, on a theoretical level, the use of "parallel" in this RSVP task would be misleading. As mentioned in my first comment, there is an alternative interpretation which would be largely serial. The word "parallel" is more typically used in the field for describing the processing of multiple stimuli shown simultaneously rather than the context of RSVP. I am worried that this would mislead readers. The data shows that at 6.7 Hz there is sufficient processing of each image to enable a classifier to tell the images apart, but not how much overlap (in a truly parallel sense) occurs.

The data that the authors present appears to show that it is really not parallel processing, but pipelining (multiple items in a sequence are processed at the same time, but not necessarily in the same processing stage at the same moment). Parallel processing as I understand it would require multiple stimuli to undergo the same processing stage at the same time (not just to be processed in the same visual system at the same time). Since each stimulus is separated by 148 ms, the current data cannot speak to parallel processing in the sense that most people would understand it.

Minor comments

p. 3, line 39, "The identification of a target stimulus involves a complex chain of brain processes that initially operate in parallel for more than 300 ms (refs 6-8)."

I assume that the authors are talking specifically about RSVP, but this needs to be more precise. Surely it is not the case that all processes active in first 300 ms are in parallel? One counter-example in vision is the two-flash threshold (Reeves, 1996). Two stimuli presented short enough in succession are reliably perceived as identical to a single pulse. Such a long parallel processing stage would also seem incompatible with auditory and tactile perception and many multisensory integration paradigms that involve segmenting sensory input on much shorter time scales (tens of milliseconds).

p. 6, line 130, "a maximum classification performance observed at 160 ms." Might this be related to the fact that a new image was shown every 148 ms? If early visual areas were somewhat serial, with a constant rate tied to the onset of the stimulus, then the impact of the subsequent image would start to arrive shortly afterwards.

p. 7, line 186, related to Figure 3A: "This pattern is similar to what was observed in the localizer task and shows that, when the stimulus was a target, the brain network activated at 150-210 ms was reactivated between 390 and 700 ms". Does it really show that the same network was reactivated? It shows that there is some overlap in the features (hence the temporal generalization) but, for example, it could be the case that only some of the features were present in both cases (for example, only the pattern in parietal cortex, but not the pattern in the visual cortex). The generalization shows that there is SOME overlap, not that the networks are identical. Indeed, if the networks were identical then the classifier performance would be as high as on the diagonal which does not appear to be the case in Figure 3 panels A or B, and the two columns in 3C are certainly similar visually but not identical.

Lines 288-297: The authors claim that RSVP disrupts feedback. However, recurrent significantly-above-chance classification performance is found on classifiers trained during a suitable time window. Would this not exactly be the behavior expected when feedback did in fact occur (the second increase

in classifiability indicating the arrival of feedback)? Especially the case that this only appears to happen on selected targets rather than non-targets seems to support this idea - the idea of feedback occurring only on targets but not non-targets is not new (e.g. Drewes, Zhu, Wutz, & Melcher, Nature: Scientific Reports 2015).

On line 361, the authors are claiming to use grayscale images of colored squares, the reasoning of which escapes me?

Reviewer #3 (Remarks to the Author):

Discrete and continuous mechanisms of temporal selection in rapid visual streams
Sebastian Marti and Stanislas Dehaene

Attentional blink – the decrease in perception immediately following presentation of a to-be-processed stimulus – has been extremely well studied at the behavioral level. Here, Marti and Dehaene provide insight into the neural mechanisms underlying the attention blink. To this end, they use MEG to record brain activity of subjects performing an RSVP task. They then use a decoding approach to investigate the dynamics of temporal selection in the brain. In particular, they put forth two different hypotheses: “gradual selection” predicts several images around the target are simultaneously selected while or “discrete selection” predicts only one image is selected at a time. Multivariate decoding analysis (MVPA) showed that both are true, but at different points in time and, potentially, in different brain regions. First, attention boosted the decodability of several stimuli, all around the time of the target. This largely occurred in visual cortices. Second, the reported target stimulus was then represented alone in frontal and parietal regions. Overall, I believe this is a strong manuscript: it is well written and the results are interesting, potentially providing new insight into the attentional blink. However, I have several significant concerns regarding the analysis and interpretation:

General Comments:

1) I have several fundamental concerns with the statistics used and their interpretation:

a) First, in several places, the authors claim differences between two different conditions based on the fact that one condition was significantly different from chance while the other condition was not. This is simply bad statistics – to claim a difference between conditions, the authors need to directly test whether there is a significant difference between the conditions. I noticed this mistake when discussing target and non-target stimuli (~line 191 and Figure 3B) and when discussing differences in classifier performance for target stimuli and nearby distractors (Figures 4C, 5B, and 5C). Such concerns are also inherent in some of the conclusions the authors draw from their cross-time decoder performances (i.e. that because it isn't significant at certain times it is therefore significantly different from other, significant, time periods).

b) The authors do not always provide statistical tests for their claims. For example, they claim there was influence of Guess 1 on Guess 2 in the behavior (paragraph starting at line 107); however, no statistical tests are provided.

c) Other times, exact p-values are not provided (i.e. sometimes just an upper-bound is provided, e.g. 'p < 0.05'). I noticed this on line 238, line 261 and line 262. Also, the p-values for all of the decoding are only reported as p < 0.05. I understand why this must be the case in the text but the p-values should be demonstrated in some form in the figures. For example, figures showing the log(p) of

decoding would give the reader a sense of the significance of the decoding (especially because the decoding levels are low).

d) The authors correct for multiple comparisons across time when using their decoder. However, it isn't clear that they correct for multiple comparisons in other ways – such as across the different conditions (Lags 1, 3, 7, and 9; e.g. in Figure 4D) and across different positions relative to the target (T-1, T-2, etc; e.g. in Figure 4A, 4B, and 4C). Did the authors also correct for these multiple comparisons? If so, how?

e) In several places I felt the authors presented data in a skewed manner. For example, line 218 states "...was above chance level for both target (...) and nearby distractors (...)". However, the T-1 distractors were not significantly above chance ($p = 0.45$) and the other two effects were modest ($p=0.02$ and 0.03). This is of particular concern because of the multiple comparison issue mentioned above. Similarly, when discussing Figure 4D in the paragraph beginning on line 224 the authors seem to have a strong positive selection bias – highlighting their desired effects but downplaying contrary effects such as decoding at position 1 during Lag-3, and completely ignoring other significant effects such as the highly significant decoding of position 12 in the Lag-3 condition. Finally, they also downplay the significant encoding of non-target stimuli shown in Figure 5C (associated text on line 255 states they were "...at chance or marginally above chance...") – as no exact p-values are provided the reader cannot evaluate the significance of these results on their own.

2) The authors take a decoding approach to tracing neural information. However, the performance of their decoder is low (<30% despite chance being 25%). In addition, the authors demonstrate different time courses for the different stimulus classes (Figure 2A). It would be useful to the reader to provide confusion matrices for the decoder – this would clarify how the decoder is failing, whether there is consistent overlap between certain categories (e.g. face and bodies) and whether this is reflecting the commonalities in time course of activation for those categories.

3) Similarly, a much more powerful decoding approach would be if the authors were able to decode specific stimuli within a category. This would alleviate concerns that the decoder is only picking up on gross activation differences between categories.

4) One of the authors central findings is that subjects select multiple stimuli early in processing but only a single stimulus later. However, it isn't clear if this is due to the nature of the task – subjects were required to report three possible targets, one at a time. Could this potentially encourage subjects to select multiple stimuli early in processing? If so, it makes sense that subjects would want those to be from the same time period in order to boost their chances of being correct. What if subjects were only allowed to report a single target stimulus – would the parallel effects disappear?

5) One of the authors central conclusions is that parallel enhancement happens in posterior visual regions while discrete enhancement occurs in frontal and parietal cortex. If this is the authors claim, then it seems they should directly test it – construct a decoder that is limited to sources from these two different sets of regions. It should show the parallel/discrete behavior. As is, the authors seem to be inferring this finding based on the timing of activation.

6) I have some concerns with how the authors present their conclusions. First, they seem to ignore a large literature on attentional blink, some of which I think their results speaks directly to (e.g. the Olivers and Meeter, Psych Rev 2008 review is very consistent with the current results!) and some of which is probably worth mentioning (e.g. literature on temporal attention, such as work from Kia Nobre, Charlie Schroeder, Gregor Thut, etc). Second, in the discussion the authors try to present their results as speaking to 'conscious processes' or 'awareness'. However, I don't see how the current

experiment addresses conscious processes in any way – in fact the stimuli that are less likely to make it into consciousness/awareness (Lag-1 and Lag-3) are just as likely to be represented in the ‘discrete manner’ (Figure 5). Finally, the authors should discuss the possibility that these results are due to temporal mis-binding between the percept of the green square (indicating the target) and the encoding of the target stimulus.

Specific comments:

1) It is not clear why in figure 4A the target category for times >350ms can not be decoded using the information at that time but can be decoded with a decoder trained from an earlier time point? For example, this is seen in the low performance of the classifier on the diagonal axis for times >350ms for stimuli T-1 and T-2 – the category identity is decodable for >350 but only if the classifier has been trained on the times <200ms.

2) Based on Figure 4D it seems that the decoder can significantly decode both T-1 and T-2 stimuli. However, this is not seen in Figure 4A. Is 4D referring to the decoder trained on earlier timepoints? It isn't clear to me from the associated text (~line 224) or the figure legend.

3) The secondary representation of the T-1 and T-2 stimuli shown in Figure 4B – when is the onset of this decoding? How does that relate to either onset of the target stimulus or the onset of sustained representation of the target (T) stimulus? From the figure it seems like it may precede the T stimulus? How is that possible?

4) Many RSVP tasks show a delay in the attentional blink such that the first couple hundred milliseconds are preserved – did the authors observe that in their current results? It seems from Figure S1 that this is not the case; why might that be different from other studies?

5) In figure 5A and 4B -- what are the vertical dotted lines? Are these the stimulus onsets?

6) The y-axis of Figure 1E seems to be mislabeled? Isn't this the variance of responses?

7) The yellow color used for Lag 7 is very hard to see (at least on my printer).

8) For Figure 4A the diagonal line does not seem to intersect the y-axis at time = 0. It seems like it should as the x-axis seems to show time relative to onset of the target stimulus (the T stimulus in green). Am I just misreading this? Or confused about the x-axis?

9) Why is there significant decoding of a stimulus well after the trial in Figure 2D? It isn't 100% clear to me which stimulus this is but this is still odd.

Response to reviewer #1:

“My first main concern derives from the choice to present the separate the items by about 148 ms in the RSVP sequence (approximately 6.7 Hz). This is long enough for the visual system to complete an initial processing of each image in a serial fashion (as the authors discuss in detail in the Discussion section). Numerous studies have provided evidence that it takes about 100-150 ms for a stimulus to get through the “first pass” of visual processing, with recurrent processing becoming increasingly important around 150-200 ms (in particular, work from people like Breitmeyer, Di Lollo or Wutz with integration masking; work on rapid identification by Thorpe and colleagues).

An alternative explanation for the same data, if it were coming, for example, from Rufin VanRullen’s lab, is that the results reflect a 7 Hz attentional sampling or “perceptual cycle” in visual processing. Note that VanRullen’s conception of sampling is actually serial, with each cycle occurring sequentially.”

This is a fundamental point raised by the reviewer. First, we would like to clarify a methodological detail of our paradigm. The stimulus onset asynchrony (SOA) in the RSVP was 116 ms with a stimulus duration of 84 ms and a inter stimulus interval of 32 ms. The frequency of presentation was thus 8.6 Hz which is typical of Attentional Blink studies using an RSVP paradigm. This has been clarified P17 §1.

Second, the referee’s is absolutely right: An important part of visual processing can be accomplished within 100-150 ms. In fact, most studies on backward masking have shown that the identification of a target stimulus is not degraded if the target-mask SOA is longer than ~100 ms even though this value can vary across experimental setup (see e.g. ¹). Regarding the present experiment, we found that stimulus processing started with successive brain responses that were systematically observed for all stimuli presented in the RSVP (Figure 2D of the manuscript). Our interpretation was that successive stimuli undergo the same “pipeline” of processes and that multiple representations can coexist but at different stages of processing. This observation is totally compatible with the idea raised by the reviewer that stimuli are serially processed, engaging the same pipeline one after the other. It is also compatible with the hypothesis of a “perceptual cycle” as proposed by Van Rullen and colleagues ². According to this framework, each stimulus in the RSVP would be integrated in a separate cycle. The present study cannot determine whether early visual processes can process multiple stimuli in parallel as it was not the main objective. This point is now mentioned in the Discussion section P13 §1.

The main goal of the study was to explore *selection* processes rather than sensory processes. Specifically, we tested whether selection processes would affect the processing of multiple stimuli in parallel (“gradual” selection) or whether stimuli would be selected one at a time (“discrete” selection). If selection processes were strictly serial, the related brain responses would not overlap in time. By contrast, the gradual selection hypothesis predicts that selection-related brain responses for multiple stimuli would overlap in time. Our experimental setup and the use of decoding algorithms allow us to tease apart these hypotheses by tracking the brain responses elicited by each stimulus in the RSVP separately from the others. We found that brain responses observed at 170 ms were reactivated later on and sustained over several hundred milliseconds when the stimulus was a target. Crucially, this was observed *simultaneously* for the target stimulus and for distractor stimuli at positions T-1 and T-2 (see Figure 4 of the manuscript). Precisely, relative to *target onset*, the stimulus at position T-2 elicited above chance classification performance between -152 and 8 ms, and

between 278 and 578 ms. Similarly for the stimulus at position T-1, we observed above-chance classification performance from -46 to 114 ms and from 354 to 724 ms. Finally, above-chance classification performance for the target stimulus was observed from 80 to 240 ms and from 410 to 690 ms (all $P_{FDR} < .05$, see Figure 4B). Thus, at least between 410 and 578 ms *after* target onset, category information for stimuli T-2, T-1 and T could be decoded simultaneously and from the same data with classifiers trained at 170 ms. This result cannot be explained by fast serial processing and strongly support the hypothesis of parallel processing in the early stages of temporal selection. This point has been clarified in the manuscript P9, last paragraph.

Future experiments could investigate whether and which early visual processes are parallel or serial. One possibility to address this question would be to manipulate the speed of the RSVP. Increasing the speed of the RSVP would allow determining which visual processing stages overlap in time. However, this goes beyond the scope of the present study which focuses on the mechanisms of temporal selection. The results revealed that both parallel and serial processing stages are observed during temporal selection but they occur at different times.

“Second, I think that the results are more confirmatory than novel. The finding that M/EEG or fMRI decoding of category can be found in the time period of around 100-300 ms and follows behavioral results has been previously shown, as the authors do a good job of pointing out in the paper. In fact, above-chance decoding also even been found for unreported (“invisible”) stimuli in other paradigms (orientation or flicker information, etc...). Thus, while it is nice to see that category decoding works at 6.7 Hz, it is not particularly surprising.

There are a number of relevant studies showing evidence for a minimum temporal duration that supports processing of a sequence of images. One nice example is Gauthier et al., J Neurosci, 2012. They varied the RSVP rate (for alternations between two images) between 33 – 4800 ms in an fMRI study. They reported that earlier visual areas responded to fast presentation rates (stimuli up to 100 – 200 ms) while later areas required longer display durations (see also McKeeff et al., 2007, showing faster rates in early visual cortex and a limit of around 100-200 ms for FFA and PPA).”

We thank the reviewer for raising this point as it allows us to clarify the novelty of the study. We fully agree with the reviewer that previous studies have already shown that the decoding of categories during RSVP is possible³. As pointed out by the reviewer, these studies revealed that the temporal tuning of areas along the ventral visual stream is not homogenous: There is a progressive slowing down of the presentation rate eliciting maximal activation from lower- to higher-order visual areas. Temporal tuning also appears to be domain-specific^{4,5}. In fact, other studies suggested the existence of multiple temporal hierarchies throughout the brain⁶.

The present study is grounded in this work but distinguish itself by a detailed exploration of temporal selection processes. The goal of this experiment was not to explore temporal tuning of the visual hierarchy. Instead, we aimed at understanding how the brain selects relevant information when it is bombarded by visual inputs. There are several important findings in our study that, we think, constitute an important contribution: First, we were able to track the processing of each stimulus in the RSVP, independently from the others, and showed that multiple representations actually coexist in sensory areas but at different stages of processing. Second, we identified which of these brain processes contribute to temporal selection and how. Third, we show that there are at least two distinct stages in temporal selection: a gradual amplification of multiple stimuli and later discrete

stages related to subjects' reports. These findings show how the computational architecture of the brain allows a rapid and efficient processing of external information, and how this information is gated to awareness.

We clarified this point in the text P12 §3.

"I was also worried that the later stage (>300 ms) does likely involve the selection of a target for response. The participant was actually supposed to respond based on the category of the target image. The design did not dissociate the decoded feature from the response, as one might have hoped. Thus, it is certainly possible that at least some aspect of the decoding performance at the later time period was about categorical response selection (linked to parietal and frontal areas) rather than about attentional selection in visual areas."

This is an important methodological aspect and two properties of our experimental design allow us to reject this hypothesis. First, during the localizer task, subjects were instructed to report the category of the stimulus – as correctly mentioned by the reviewer – but the mapping between the category and the motor response was systematically shuffled on each trial so that subjects could not anticipate their motor response before the appearance of the response screen. Second, during the dual-task (RSVP), subjects gave their response by orally naming the letters corresponding to the images they selected. Again, the order of the images on the response screen was randomly shuffled on each trial. Hence, it was impossible for subjects to anticipate their response before the presentation of the response screen.

Therefore, the brain responses revealed by classifiers trained at the later time period (>300 ms) cannot be related to the response selection and can only reflect stimulus selection processes.

This has been clarified P16 §3, P17 §2 of the main manuscript.

"My third concern is the more general theoretical implications. The authors do not really motivate why performance in an RSVP task is so important and I found the discussion of RSVP to lack precision. In terms of theory, in the real world, natural stimuli are not very often presented at an RSVP rate of 6.7 Hz, but rather much faster (video) or slower (saccades at 3-5 Hz). I understand that this is a short format, but it is left open what is actually processed (gist recognition) at this rate. On the one hand, it is claimed that it is efficient at 10 Hz, but then claimed that the brain cannot fully process them (whatever that means), and some images are subjectively invisible. But the studies cited cover a wide variety of stimuli, including both words and photographs, and differing amount of masking. Holcombe (2009), for example, has argued for a cut-off at around 10 Hz."

The motivation of using an RSVP rate around 10 Hz is that interesting behavioral effects are observed. Past studies showed that at this rate, participants are capable of identifying a target stimulus with a fairly good performance, but when their attention is distracted by another task (typically another target stimulus in the stream) the selection performance is incredibly degraded⁷. During a short and specific period of time (~half a second), the selection processes are severely disrupted and almost any stimulus in the stream can be substituted to the target⁸. The question of what is actually processed at this presentation rate still remains. Our experimental approach allows us to answer this question at least in part. First, in the dual-task experiment, subjects were asked to identify specifically the target image (rather than reporting its category). Their performance indicate that they successfully achieve this task: they correctly reported the target stimulus as their first

choice in most of the trials. If subjects were only capable of reporting images at some level of abstraction (e.g. a face rather than an object) without being able to precisely identify it, their performance would be much lower: The chance to select the target stimulus while only perceiving the category is 1/10 (there were 10 possible stimuli within each category). Thus, subjects' behavior suggests that they were able to fully process target stimuli. Second, at the brain level, the amplitude of late responses (>350 ms) identified with decoding techniques reflected the subjects' order of preference: the stimuli selected as subjects' first guess elicited the strongest late brain responses. This shows that brain responses to specific stimuli in the RSVP (rather than brain responses to general categories) were related to subjects' behavior. These results indicate that although our task was challenging, the brain was able to fully process target stimuli but this ability partly relied on attentional resources. It is however impossible from these data to determine with precision what information was extracted for the other, irrelevant stimuli presented in the RSVP.

We have clarified our description of subjects' performance in RSVP tasks P3 §1 and P13 §2.

"I would be worried that, on a theoretical level, the use of "parallel" in this RSVP task would be misleading. As mentioned in my first comment, there is an alternative interpretation which would be largely serial. The word "parallel" is more typically used in the field for describing the processing of multiple stimuli shown simultaneously rather than the context of RSVP. I am worried that this would mislead readers. The data shows that at 6.7 Hz there is sufficient processing of each image to enable a classifier to tell the images apart, but not how much overlap (in a truly parallel sense) occurs.

The data that the authors present appears to show that it is really not parallel processing, but pipelining (multiple items in a sequence are processed at the same time, but not necessarily in the same processing stage at the same moment). Parallel processing as I understand it would require multiple stimuli to undergo the same processing stage at the same time (not just to be processed in the same visual system at the same time). Since each stimulus is separated by 148 ms, the current data cannot speak to parallel processing in the sense that most people would understand it."

We fully agree with the reviewer that the early processes between ~100 and ~170 ms after stimulus onset and observed for all items in the RSVP (Figure 2D of the manuscript) could be conceived as a pipeline of processes. The present data do not provide any evidence that these stages of processing are able to process multiple stimuli simultaneously. However, they do show that these processes can be deployed in parallel to Task 1: significant decoding of categories was observed for each stimulus in the RSVP, independently of their position in the RSVP. This is precisely what we meant when using the term parallel. Previous studies have reported similar findings indicating that early visual processes operate even if subjects are focused on another task⁹⁻¹². In this specific context, and in accordance with previous dual-task studies on similar topics, we think the term parallel is appropriate to characterize these processes. We have rephrased our interpretation in the manuscript P13 §1 in order to avoid any confusion.

Furthermore, as we already mentioned in response to the reviewer's point #1, another part of our results indicate parallel processing. Classifiers trained at 170 ms revealed that the sustained activity observed for three successive stimuli presented at positions T-2, T-1 and T largely overlapped in time, indicating that these stimuli were processed at the same stage in parallel. This point is also clarified in the text P14 §1.

“p. 3, line 39, “The identification of a target stimulus involves a complex chain of brain processes that initially operate in parallel for more than 300 ms (refs 6-8).” I assume that the authors are talking specifically about RSVP, but this needs to be more precise. Surely it is not the case that all processes active in first 300 ms are in parallel? One counter-example in vision is the two-flash threshold (Reeves, 1996). Two stimuli presented short enough in succession are reliably perceived as identical to a single pulse. Such a long parallel processing stage would also seem incompatible with auditory and tactile perception and many multisensory integration paradigms that involve segmenting sensory input on much shorter time scales (tens of milliseconds).”

This sentence specifically applies to a dual-task situation where two tasks are performed independently of each other. Essentially, we found in this study that the successive brain responses deployed in the first ~300 ms are not affected by the presence of a preceding task. We have modified the sentence accordingly P3 §2.

“p. 6, line 130, “a maximum classification performance observed at 160 ms.” Might this be related to the fact that a new image was shown every 148 ms? If early visual areas were somewhat serial, with a constant rate tied to the onset of the stimulus, then the impact of the subsequent image would start to arrive shortly afterwards.”

The maximum classification performance was observed at ~160 ms both in localizer and in dual-task sessions (see Figures 2B-C, and S4). Given that, during the localizer, only one image was presented in each trial, the peak of the classification performance cannot be due to a subsequent image.

“p. 7, line 186, related to Figure 3A: “This pattern is similar to what was observed in the localizer task and shows that, when the stimulus was a target, the brain network activated at 150-210 ms was reactivated between 390 and 700 ms”. Does it really show that the same network was reactivated? It shows that there is some overlap in the features (hence the temporal generalization) but, for example, it could be the case that only some of the features were present in both cases (for example, only the pattern in parietal cortex, but not the pattern in the visual cortex). The generalization shows that there is SOME overlap, not that the networks are identical. Indeed, if the networks were identical then the classifier performance would be as high as on the diagonal which does not appear to be the case in Figure 3 panels A or B, and the two columns in 3C are certainly similar visually but not identical.”

This is absolutely correct. The manuscript has been modified accordingly P6 §3 and P8 §3.

“Lines 288-297: The authors claim that RSVP disrupts feedback. However, recurrent significantly-above-chance classification performance is found on classifiers trained during a suitable time window. Would this not exactly be the behavior expected when feedback did in fact occur (the second increase in classifiability indicating the arrival of feedback)? Especially the case that this only appears to happen on selected targets rather than non-targets seems to support this idea - the idea of feedback occurring only on targets but not non-targets is not new (e.g. Drewes, Zhu, Wutz, & Melcher, Nature: Scientific Reports 2015).”

We completely agree with the reviewer. The paragraph was ambiguous. We meant that feedback attentional signals were disrupted for *irrelevant, non-target* stimuli in the RSVP. The point of this paragraph is that the ability of the brain to quickly extract complex information such as category or meaning from rapidly changing visual inputs seems to rely on feedforward processes. The paragraph has been corrected accordingly P13 §2.

"On line 361, the authors are claiming to use grayscale images of colored squares, the reasoning of which escapes me?"

Corrected P16 §2.

Responses to reviewer #2:

"1) I have several fundamental concerns with the statistics used and their interpretation:

We would like to thank the reviewer for his/her insightful comments. We have made a special effort to (i) clarify the approach required to address the questions of interest, (ii) provide all needed statistical information, including not significant results, (iii) improve the description of the results with more details to avoid any bias in our interpretation, (iv) add detailed (yet synthetic) statistical information in the figures. Below, we provide detailed responses to each point mentioned by the reviewer.

a) First, in several places, the authors claim differences between two different conditions based on the fact that one condition was significantly different from chance while the other condition was not. This is simply bad statistics – to claim a difference between conditions, the authors need to directly test whether there is a significant difference between the conditions. I noticed this mistake when discussing target and non-target stimuli (~line 191 and Figure 3B) and when discussing differences in classifier performance for target stimuli and nearby distractors (Figures 4C, 5B, and 5C). Such concerns are also inherent in some of the conclusions the authors draw from their cross-time decoder performances (i.e. that because it isn't significant at certain times it is therefore significantly different from other, significant, time periods)."

We agree with the reviewer. We have added the missing direct comparisons between the relevant conditions, e.g. target vs non-target, or target vs nearby distractors. However, in the context of the present experiment, a direct comparison between conditions is not sufficient. To take an example, it could be that both target and non-target stimuli induced a sustained activity over a long period of time, but target stimuli induced stronger responses. The interpretation of this finding would be that even though one item induces stronger activations than the other, target and non-target stimuli still undergo the same stages of processing. In order to determine whether the processing of target and non-target stimuli are qualitatively different, i.e. that certain processing stages are observed in one condition but not in the other, we also need to compare the classification performance within conditions to chance level. Therefore, when relevant, we performed both a direct comparison between conditions and a comparison with chance level. The relevant parts of the manuscript have been modified:

- Localizer vs RSVP: P7 §2.
- Target vs non-target: P8 §2.
- Target vs nearby distractors: P10 §2; 11 §3.
- Effect of inter target lag: P10 §3; 11 §4
- Effect of subjects' guesses: P10 §4; 11 §5

"b) The authors do not always provide statistical tests for their claims. For example, they claim there was influence of Guess 1 on Guess 2 in the behavior (paragraph starting at line 107); however, no statistical tests are provided."

Exact P-values are now indicated for all statistical tests unless it was below .001. We also include tests that were not significant.

"c) Other times, exact p-values are not provided (i.e. sometimes just an upper-bound is provided, e.g. 'p < 0.05'). I noticed this on line 238, line 261 and line 262. Also, the p-values for all of the decoding are only reported as p < 0.05. I understand why this must be the case in the text but the p-values should be demonstrated in some form in the figures. For example, figures showing the log(p) of decoding would give the reader a sense of the significance of the decoding (especially because the decoding levels are low)."

As we mentioned previously, exact p-values are now indicated. In addition, in order to provide the relevant statistical information in the figures (Figures 2B-C, 4B, 5A) without overloading them, we chose to represent the log(p) values, as suggested by the reviewer, by a color code with darker colors representing lower p-values.

"d) The authors correct for multiple comparisons across time when using their decoder. However, it isn't clear that they correct for multiple comparisons in other ways – such as across the different conditions (Lags 1, 3, 7, and 9; e.g. in Figure 4D) and across different positions relative to the target (T-1, T-2, etc; e.g. in Figure 4A, 4B, and 4C). Did the authors also correct for these multiple comparisons? If so, how?"

Following the reviewer's recommendation, we modified the way we applied corrections for multiple comparisons: unless otherwise specified, FDR correction was applied across testing times, specific training times (120, 170, 220, 270, 320, 370 ms) and conditions (e.g. target vs non-target, inter target lag or stimulus position). The P-values reported in the manuscript have been modified accordingly (results section P4-11). Additional precisions are provided in the method section P19 last paragraph, and specific information are indicated in the figure captions.

"e) In several places I felt the authors presented data in a skewed manner. For example, line 218 states "...was above chance level for both target (...) and nearby distractors (...)". However, the T-1 distractors were not significantly above chance (p = 0.45) and the other two effects were modest (p=0.02 and 0.03). This is of particular concern because of the multiple comparison issue mentioned above. Similarly, when discussing Figure 4D in the paragraph beginning on line 224 the authors seem to have a strong positive selection bias – highlighting their desired effects but downplaying contrary effects such as decoding at position 1 during Lag-3, and completely ignoring other significant effects such as the highly significant decoding of position 12 in the Lag-3 condition. Finally, they also downplay the significant encoding of non-target stimuli shown in Figure 5C (associated text on line 255 states they were "...at chance or marginally above chance...") – as no exact p-values are provided the reader cannot evaluate the significance of these results on their own."

We thank the reviewer for pointing this out. This part of the manuscript has been completely re-worked and simplified. The goal of these analyses was to determine whether target-selective brain responses (i) depends on attentional resources and (ii) are related to subjects' behavior. We computed the mean classification performance between 400 and 550 ms after stimulus onset for stimuli at target positions. A repeated-measure ANOVA on aligned rank transformed data was then conducted with the inter target lag and subjects' guesses as within-subjects factors. The results of this analysis are described in details P10 §3-4, P11 §4-5, and P12 §1.

Briefly, for early brain responses (i.e. at a training time of 170 ms), we found a significant effect of inter target lag. The classification performance exhibited a U-shape curve with the lowest performance observed for lag 3 condition (Figure 4D). The main effect of Guesses did not reach significance, although the classification performance was above chance and highly similar for Guess 1 and 2 but not for Guess 3 and unreported stimuli. No significant interaction was observed between these two factors. These results suggest that early sustained responses reflected attentional modulations of category-selective brain activity but were only partially related to subjects' behavior as it did not reflect subjects' order of preference.

The same analysis was conducted for late brain responses (i.e. at a training time of 370 ms). We found a main effect of inter target lag and a main effect of Guesses but no interactions between these factors. The classification performance at lag 1 was lower than in the other lags. Regarding the effect of Guesses, we found that classification performance linearly increased from unreported stimuli to stimuli reported as Guess 1. This shows that the multiple guesses produced by subjects were differentiated mainly by the amplitude of late brain responses.

"2) The authors take a decoding approach to tracing neural information. However, the performance of their decoder is low (<30% despite chance being 25%). In addition, the authors demonstrate different time courses for the different stimulus classes (Figure 2A). It would be useful to the reader to provide confusion matrices for the decoder – this would clarify how the decoder is failing, whether there is consistent overlap between certain categories (e.g. face and bodies) and whether this is reflecting the commonalities in time course of activation for those categories."

The classification performance in our experiment is indeed low. This is understandable since the MEG signal is an indirect noisy measure of brain activity. In addition, we specifically focused on attentional modulations which are known to be of modest amplitude but sustained over time. This has been observed in multiple studies of attention in monkeys^{13, 14} and humans^{15, 16}. However, as can be seen in Figure 2B of the manuscript, the noise in our experiment is also very low. For instance, the standard deviation across subjects during the baseline period in the localizer task is $SD = .00014$. The decoding performance reflecting the attentional modulation observed for classifiers at 150-200 ms was low ($M = .257$, $SEM = .0016$, average over a time window ranging from 400 to 550 ms) but it was still much higher than baseline ($M = 52 SD$). Thus, even though the classification performance was overall low, our measures were clearly sensitive enough to reveal the effects of interest.

Furthermore, following the reviewer's suggestion, we computed confusion matrices to examine the classification time course for each category. As can be seen in Figure S4A of the manuscript, classifiers were specific to a category and no systematic overlap between categories was observed. For instance, classifiers for faces at 150-200 ms accurately classified face stimuli on 47 % of the trials but performed poorly on other categories (16, 19 and 18 % respectively for place, object and body parts stimuli). Furthermore, there was a difference in the performance of the classifiers. For instance, during the same time period, classifiers for places, objects and body parts performed at 43, 32 and 38 % respectively (Main effect of stimulus category: $F(3,42) = 20.55$, $P < .001$).

Importantly, classifiers for all four categories had similar time courses even though the overall decoding performance varied across categories (Figure S4B). Category-specific time courses had similar onsets and offsets (defined as the first and last points exceeding the 50th percentile of the distribution measured on group averaged data): 110-470, 90-450, 120-450, and 90-500 ms

respectively for face, place, object and body parts categories. This shows that the classification performance averaged across categories, as reported in the main manuscript, faithfully reflected overall category-selective responses in the brain.

The confusion matrices are now included in the manuscript (Figure S4) and discussed P6 §2.

"3) Similarly, a much more powerful decoding approach would be if the authors were able to decode specific stimuli within a category. This would alleviate concerns that the decoder is only picking up on gross activation differences between categories."

This is an interesting approach. We applied the same decoding procedure as for categories (see the Method section of the manuscript) except that the label given to the classifier were single stimuli. Classifiers were trained on data from the localizer task and then applied to the RSVP data. In both tasks, the identity of single stimuli could be decoded from MEG data from 90 to ~250 ms (Figure R1A, all $P_{\text{FDR}} < .05$, chance = .025). Temporal generalization revealed a diagonal shape, suggesting that the processing of stimulus identity essentially consisted in a series of brief processing stages. The strongest response was observed for classifiers trained at 170 ms which performed above chance between 100 and 200 ms with a peak at $M = .028$.

Figure R1B shows the overlay of the temporal generalization matrices for each of the 12 stimuli following T1. The results were not as strong as what was observed with category-classifiers. The classification performance was significantly above chance for all stimuli only when considering uncorrected P-values, as shown in Figure R1B. Still, it seems possible to decode the identity of every single stimulus presented in the RSVP. Furthermore, the processing of each stimulus appears again highly systematic, starting at ~90 ms and ending around ~250 ms. This suggests that stimuli induced a series of processes, possibly in the form of a pipeline.

Figure R1. Decoding of single stimuli. (A) Performance of classifiers trained to separate single stimuli in the localizer task and applied to the RSVP task. Top graph: One classifier was trained at each time sample and tested on the same time sample. The shaded grey area represents the standard error to the mean across subjects. Right graphs: Classifiers were trained at specific time samples (from 120 to 370 ms) and tested on all other time samples. Results from signed rank tests comparing classification performance to chance are represented by the thick line below the X axis with darker colors representing lower P-values (uncorrected for multiple comparisons). Matrix plot: Classifiers were systematically trained on each time sample and tested on all others. The color code represents the classification performance and the dotted line the diagonal of the matrix. (B) Contour plot representing classification performance significantly above chance (corrected for multiple comparisons) obtained for each stimulus and averaged across subjects. Colors represent the successive stimuli in the RSVP (except T1) depicted below by a small rectangle. Dotted lines represent the diagonals of each matrix plot. (C) The panels represent temporal generalization matrices for the target stimulus (T) and distractors at positions T-4 to T+2 averaged across lag 7 and 9 conditions. (D, E) Classification performance as a function of time for a classifier trained at 170 (D) and 370 ms (E) after stimulus onset. Filled areas represent time samples at which the classification performance was significantly different from chance

(Signed-rank tests, $\alpha = .05$, FDR corrected for multiple comparisons across stimulus positions, training (170, 370 ms), and testing times). Colored dotted lines represent stimuli onsets.

We next investigated the effect of the temporal proximity of the target on nearby distractors. Although the classification performance exceeded chance, almost none of these were significant after correction for multiple comparisons. Still, two important differences with category classification should be noticed. First, classifiers trained at 170 ms revealed a sustained activity between 400 and 550 ms only after the presentation of a target stimulus (Figure R1C-D, $W = 104$, $P_{\text{uncorr}} = .01$). The classification performance was at chance for all other stimuli (all $P_{\text{uncorr}} > .3$). This suggests that a biphasic response was observed only for the target stimulus, unlike what was observed when using category-selective classifiers. Second, we found no trace of late brain responses. The performance of classifiers trained at 370 ms was at chance (Figure R1E). The average classification performance between 400 and 550 ms was not different from chance for any of the stimuli, even when considering uncorrected P-values (all $P_{\text{uncorr}} > .25$). Therefore, although these analyses should be taken with caution because of the low statistical power, they suggest that only the identity of the target stimulus was maintained.

These results suggest that it is possible to some extent to decode the identity of single stimuli presented in an RSVP but the poor quality of the results obtained suggest that the current experimental design might not be optimal for this. A first reason is that there is probably not enough trials for the training of the classifiers. Subjects performed 300 trials of the localizer task which leaves only 7 to 8 examples of each stimulus. With such a low number of examples by class, the SVM might not be able to find the optimal hyperplane that best separate the classes. The classifier would thus perform poorly when generalized to unseen examples of a given class. A second reason is that it is most likely that stimulus-specific classifiers use low-level visual features primarily encoded in early areas such as V1 or V2. Previous studies showed that the effects of attention are much weaker in early sensory areas compared to higher-order areas¹⁴. It is thus plausible that the methods used in the present study were not sensitive enough to capture the attentional effects that might occur in early visual areas.

In conclusion, it seems that our paradigm is not well adapted to decode the identity of unique stimuli. The experiment was designed to make use of the well characterized brain responses to specific categories in order to track the activations elicited by each stimulus in the RSVP. The results showed that this approach was successful. Not only we were able to separate these overlapping brain responses but we also identified the attentional modulations induced by target stimuli. Therefore, we think that, in the current context, the decoding of categories is the most appropriate approach to track brain responses to single stimuli presented in RSVP.

“4) One of the authors central findings is that subjects select multiple stimuli early in processing but only a single stimulus later. However, it isn’t clear if this is due to the nature of the task – subjects were required to report three possible targets, one at a time. Could this potentially encourage subjects to select multiple stimuli early in processing? If so, it makes sense that subjects would want those to be from the same time period in order to boost their chances of being correct. What if subjects were only allowed to report a single target stimulus – would the parallel effects disappear?”

This is a very interesting question. It is possible that the parallel enhancement observed in the early stages of selection is due to the task itself. We agree that if subjects were asked to report only one

target, this effect could be reduced or could even completely disappear. This hypothesis is not in contradiction with our findings. Our results show that when the brain is bombarded by visual inputs, the selection processes are not restricted to serial, discrete mechanisms. Instead, attention can spread in time over multiple stimuli and affect them simultaneously. Following the reviewer's idea, it could be that the attentional time window shrinks or expands in time, as it has been observed in space¹⁷, depending on the goal of the subject. This is a very stimulating idea that could be addressed in future experiments. We now mention this point in the discussion section P14 §1.

"5) One of the authors central conclusions is that parallel enhancement happens in posterior visual regions while discrete enhancement occurs in frontal and parietal cortex. If this is the authors claim, then it seems they should directly test it – construct a decoder that is limited to sources from these two different sets of regions. It should show the parallel/discrete behavior. As is, the authors seem to be inferring this finding based on the timing of activation."

First, we would emphasize that our main objective was to test the existence of parallel and/or discrete mechanisms during temporal selection. We don't want to make a strong claim regarding the precise spatial location of these processes as it is obviously limited by the spatial resolution of the MEG.

Although limited, we see mainly two different approach to provide spatial information. One is to localize the brain responses used by the classifiers to separate the classes. This is the method we describe in the manuscript. We provide the location of the informative brain activity reconstructed from the SVM weights. Figure 3C shows the location of informative brain responses for classifiers trained at 150-200 ms and 350-400 ms. In the early time window, most of the informative activity was located in visual areas for all categories. By contrast, in the late time window, informative activity was located again in the visual areas but also in other areas such as the parietal cortex and to a minor extent the frontal cortex (e.g. precentral areas).

The other approach is the one proposed by the reviewer: to decode the stimulus categories from specific regions of interest in source space. In the following, we present the results of this analysis. Specific regions of interest in frontal (superior, middle and precentral areas), parietal (superior and inferior lobules) and visual areas (lingual, lateral occipital, fusiform and parahippocampal areas, pulled together and considered as a single ROI) were a priori selected based on the Desikan-Killiany atlas¹⁸. Activations in these ROIs were extracted for each trial and a linear SVM was trained to separate stimulus categories separately for each ROI, following the same procedure as in the sensor space. The hypothesis at test is that gradual processing stages would be mainly observed in visual areas while late discrete stages would mainly rely on higher-order areas such as the parietal and the frontal cortex.

The results are depicted in Figure R2. As can be seen, stimulus-categories could be decoded from all ROIs. Importantly, for all of them, classifiers trained at 170 ms exhibited a biphasic response with a first sharp peak between ~100 and ~200 ms and a sustained phase between ~290 and the end of the epoch (all $P_{FDR} < .05$). Similarly, classifiers trained at 370 ms induced sustained activity from ~240 ms to the end of the epoch (all $P_{FDR} < .05$). Finally, a direct comparison of the classification performance between ROIs did not reveal any significant differences (average classification performance between 400 and 550 ms, all $P_{FDR} > .38$). Thus, there was no evidence that early gradual and late discrete stages of processing relied on different brain areas.

This approach is however limited by the spatial precision of the MEG. The signal resulting from the neuronal activity is typically captured by multiple MEG sensors. Even when projected in source space, the activity in a given brain area might be contaminated by the activity of another, distant area. It is probable that even if a classifier is trained on a specific ROI, it might be able to extract information originating from a distant brain area. The fact that classifiers trained on different ROIs had similar performance profiles does not necessarily imply that all these ROIs performed the same computation. It is therefore difficult to conclude from these results.

To conclude, we chose to keep the approach originally described in the manuscript, which essentially consists in visualizing the weights of the SVM in source space in order to provide the location of the informative brain activity. However, we also modified the manuscript in order to tone down our interpretation of the spatial location of parallel and discrete processes P9 §1.

Figure R2. Decoding stimulus categories in source space. Each panel represents the temporal generalization matrices (left) obtained for classifiers trained to separate stimulus categories from activations in specific ROIs. The time courses of classifiers trained at 170 (middle) and 370 ms (right) are also depicted. The location of the ROIs are represented by colored surfaces on the lateral view of the left hemisphere and the medial view of the right hemisphere. ROIs include the superior (light blue), middle (blue) and precentral (dark blue) frontal areas, the superior (orange) and inferior (red) parietal lobules, the lingual, lateral occipital, fusiform and parahippocampal visual areas (green). Results from signed rank tests comparing classification performance to chance are represented by areas filled in grey. Darker colors representing lower P-values. FDR correction for multiple comparisons was applied across specific training times (120 to 370 ms), testing times and ROIs.

“6) I have some concerns with how the authors present their conclusions. First, they seem to ignore a large literature on attentional blink, some of which I think their results speaks directly to (e.g. the Olivers and Meeter, Psych Rev 2008 review is very consistent with the current results!) and some of which is probably worth mentioning (e.g. literature on temporal attention, such as work from Kia Nobre, Charlie Schroeder, Gregor Thut, etc). Second, in the discussion the authors try to present their results as speaking to ‘conscious processes’ or ‘awareness’. However, I don’t see how the current experiment addresses conscious processes in any way – in fact the stimuli that are less likely to make it into consciousness/awareness (Lag-1 and Lag-3) are just as likely to be represented in the ‘discrete manner’ (Figure 5). Finally, the authors should discuss the possibility that these results are due to temporal mis-binding between the percept of the green square (indicating the target) and the encoding of the target stimulus.”

We thank the reviewer for pointing out this highly relevant literature. Indeed the “Boost and Bounce” model of the AB is relevant to the present study, particularly in the context of the “lag-1 sparing” phenomenon: The AB typically reaches its maximum at lags 2-3 (i.e. 200-300 ms after the first target) but surprisingly subjects’ performance recovers at the shortest lag (‘lag-1 sparing’). It has been debated whether at lag-1 the first and the second target are perceived serially as two separate events, or in parallel as a single perceptual event, which would imply that multiple stimuli can be consciously perceived simultaneously. The Boost and Bounce model suggests that the target stimulus triggers attentional enhancement which affects not only the target but also the successive stimuli presented at the same location¹⁹. The consolidation of T2 in working memory can only proceed once the attentional set switches from T1 to T2. Therefore, T2 can benefit from attentional enhancement if it shares features with T1, but its consolidation is nevertheless delayed. According to the model, the attention devoted to T2 at lag-1 is sufficient to explain why its perception is spared. Interestingly, another dominant model of the AB the “simultaneous type, serial token” (ST²) model²⁰ developed by Wyble and colleagues, makes a different prediction. After an initial attentional enhancement, T1 and T2 stimuli enter awareness simultaneously if T2 is presented at lag-1. T2 perception is spared at lag-1 not only because it receives attentional signals but also because T1 and T2 access the same conscious episode. Regarding the present experiment, the ST² model would predict that late responses to temporally close stimuli should overlap in time while the boost and bounce model makes the opposite prediction. Furthermore, ST² predicts that T2 late responses at lag-1 should be strong while the Boost and Bounce model predicts a degradation induced by the collision with T1. The present results are compatible with the Boost and Bounce model and support the serial view: multiple consecutive stimuli benefit from attentional enhancement, but they access awareness only in a discrete manner. This suggests that at lag 1, T2 would benefit from the attention devoted to T1, but conscious access would still be delayed. We now discuss computational models of the attentional blink, including the Boost and Bounce model and the ST² model P15 §1 of the manuscript. We also cite relevant work by K. Nobre and C. Schroeder P3 §1.

In a second point, the reviewer mentioned that the link between our results and conscious processing was unclear. First, we would like to clarify what we mean by ‘conscious’: In the context of the current experiment, we specifically refer to subjects’ reports. Subjects systematically produced three reports at the end of each trial. We assume that they reported what they were conscious of during a trial. Second, the fact that late ‘discrete’ brain responses were observed at short lags, where target stimuli are less likely to be perceived, as well as at long lags does not necessarily imply that it was observed equally for conscious and non-conscious stimuli. Indeed, the decoding analyses revealed that there was a link between discrete brain responses and subjects’ conscious report: it was observed for stimuli reported as Guess 1 and 2 but not for Guess 3 and unreported stimuli. Furthermore, the classification performance predicted the subjects’ order of preference (Guess 1, 2 and 3). This shows that ‘discrete’ brain responses were related to either the conscious representation of the stimulus or its gating to awareness. We clarified this point in the manuscript P14 §2.

Finally, the reviewer suggested that the present results could be explained by a mis-binding between the target cue (the green square) and the stimulus. This interpretation is completely valid. The presentation of the green square should trigger the orientation of temporal attention toward the image. Error trials in which the target stimulus is substituted by a distractor could result from the inability to bind the target feature to the image. At short lag, the lack of attentional resources could

increase the probability of mis-binding. This hypothesis appears complementary to our interpretation of the results and is now discussed P14 §3.

Specific comments:

1) It is not clear why in figure 4A the target category for times >350ms can not be decoded using the information at that time but can be decoded with a decoder trained from an earlier time point? For example, this is seen in the low performance of the classifier on the diagonal axis for times >350ms for stimuli T-1 and T-2 – the category identity is decodable for >350 but only if the classifier has been trained on the times <200ms.

The classifiers were trained in the localizer task. During this task, classifiers trained at late latencies (>350 ms) were able to extract category-related information (see Figure 2B of the manuscript). This indicates that there was a specific pattern of activity that differentiated the two conditions. Once these classifiers were generalized to the RSVP task, they performed at chance. Since SVM classifiers are spatially selective, this does not mean that there was no activity at all, but only that the classifiers could not discriminate the conditions based on the measured activity. The fact that classifiers trained at 170 ms performed steadily above chance beyond 350 ms shows that the activity during this time period was essentially a prolongation or a reactivation of earlier patterns. This was accompanied by additional late stages of processing for target but not for distractor stimuli.

2) Based on Figure 4D it seems that the decoder can significantly decode both T-1 and T-2 stimuli. However, this is not seen in Figure 4A. Is 4D referring to the decoder trained on earlier timepoints? It isn't clear to me from the associated text (~line 224) or the figure legend.

Figure 4D has been changed. It shows the classification performance averaged over 400-550 ms for classifiers trained at 170 ms. This information has been added to the manuscript P10 §2 and to the figure legend.

3) The secondary representation of the T-1 and T-2 stimuli shown in Figure 4B – when is the onset of this decoding? How does that relate to either onset of the target stimulus or the onset of sustained representation of the target (T) stimulus? From the figure it seems like it may precede the T stimulus? How is that possible?

For stimulus at position T-2, there were two time windows during which the classification performance was above chance. Relative to *target onset*, the stimulus at position T-2 elicited above chance classification performance between -152 and 8 ms, and between 278 and 578 ms. Similarly for stimulus at position T-1, we observed above-chance classification performance from -46 to 114 ms and from 354 to 724 ms. Finally, above-chance classification performance for the target stimulus was observed from 80 to 240 ms and from 410 to 690 ms (all $P_{FDR} < .05$, see Figure 4B). Hence, the secondary representation of stimuli at positions T-1 and T-2 occurred *after* the onset of the target stimulus, and temporally overlapped with the secondary representation of the target stimulus.

We added this precision in the text to avoid any confusion (P9, last paragraph). In addition, we noticed a mistake in the time axis of Figure 4B. This has been corrected.

4) Many RSVP tasks show a delay in the attentional blink such that the first couple hundred milliseconds are preserved – did the authors observe that in their current results? It seems from Figure S1 that this is not the case; why might that be different from other studies?

This is a very interesting point. Many studies of the Attentional Blink report a preserved T2 identification performance when the stimulus is presented within 100-150 ms after T1, the so-called ‘lag-1 sparing’. This phenomenon has interesting properties, and we would like to remind some of these here. First, a lag-1 sparing is not systematically observed in AB studies. Rather, it seems to depend on the paradigm used. For instance, lag-1 sparing is removed if task-settings are changed between T1 and T2 while the AB itself remains unchanged²¹. Second, there is evidence that the amplitude of lag-1 sparing is unrelated to the amplitude of the AB²². Third, its amplitude is influenced by bottom-up attentional filtering²³, the size of the focus of attention²⁴ and to a minor extent top-down control²³.

Regarding the current study, the group average reported in Figure S1 indicates that there is no reliable lag-1 sparing. However, a detailed examination of single-subjects data shows that this results from a mixture of subjects with and subjects without lag-1 sparing. To estimate the size of lag-1 sparing in each subject, we subtracted T2 recognition performance at lag 1 and at lag 3, where the AB is supposed to reach its maximum. Eight subjects over 15 had a lag-1 sparing above 0 but of moderate size (Figure R3A). The overall performance for subjects without a lag-1 sparing was lower (averaged T2 recognition performance at lag 7 and 9: $M = 72\%$) compared to subjects with a lag-1 sparing ($M = 86\%$; $t(13) = 2.67$, $P = .02$). The size of lag-1 sparing was correlated with the averaged task performance at long inter target lag (Spearman’s rho = .57, $P = .03$) but not with the size of the AB (defined as the difference in performance between lag 9 and lag 3 conditions: Spearman’s rho = .075, $P = .79$).

Figure R2: (A) Size of the lag-1 sparing for each subject. The lag-1 sparing is defined as the difference in performance between lag 1 and lag 3 conditions. Red and black dots represent subjects with and without lag-1 sparing (i.e. superior or equal to zero) respectively. (B) Averaged proportion of correct T2 identification (\pm s.e.m.) as a function of inter target lag for subjects with (red) and without (black) lag-1 sparing.

These results show that although subjects’ performance had similar profile at lag 3, 7 and 9, two patterns emerged at lag 1. Some subjects exhibited the typical U-shape while for others the performance linearly increased with the inter target lag. It appears that the presence or absence of lag-1 sparing in AB studies mostly depends on the paradigm used and on individual performance. The lag-1 sparing constitutes a research question on its own and this goes beyond the scope of the present study. Future studies should explore how subjects’ performance and lag-1 sparing are related.

5) In figure 5A and 4B -- what are the vertical dotted lines? Are these the stimulus onsets?

Yes. This is now indicated in the captions.

6) *The y-axis of Figure 1E seems to be mislabeled? Isn't this the variance of responses?*

Corrected.

7) *The yellow color used for Lag 7 is very hard to see (at least on my printer).*

Yellow and green colors for lag 7 and 9 have been replaced by red and orange colors respectively in figures 1 and 2, S2.

8) *For Figure 4A the diagonal line does not seem to intersect the y-axis at time = 0. It seems like it should as the x-axis seems to show time relative to onset of the target stimulus (the T stimulus in green). Am I just misreading this? Or confused about the x-axis?*

The time axis was erroneously locked to T1 onset. This has been corrected.

9) *Why is there significant decoding of a stimulus well after the trial in Figure 2D? It isn't 100% clear to me which stimulus this is but this is still odd.*

This figure has been updated. It's now showing above-chance classification performance corrected for multiple comparison across training time, testing time and stimuli.

References

1. Enns, J.T. & Di Lollo, V. What's new in visual masking? *Trends Cogn Sci* **4**, 345-352 (2000).
2. VanRullen, R. & Koch, C. Is perception discrete or continuous? *Trends Cogn Sci* **7**, 207-213 (2003).
3. Gauthier, B., Eger, E., Hesselmann, G., Giraud, A.L. & Kleinschmidt, A. Temporal tuning properties along the human ventral visual stream. *J Neurosci* **32**, 14433-14441 (2012).
4. Stigliani, A., Weiner, K.S. & Grill-Spector, K. Temporal Processing Capacity in High-Level Visual Cortex Is Domain Specific. *J Neurosci* **35**, 12412-12424 (2015).
5. Yeatman, J.D. & Norcia, A.M. Temporal Tuning of Word- and Face-selective Cortex. *Journal of cognitive neuroscience* **28**, 1820-1827 (2016).
6. Chaudhuri, R., Knoblauch, K., Gariel, M.A., Kennedy, H. & Wang, X.J. A Large-Scale Circuit Mechanism for Hierarchical Dynamical Processing in the Primate Cortex. *Neuron* **88**, 419-431 (2015).
7. Raymond, J.E., Shapiro, K.L. & Arnell, K.M. Temporary suppression of visual processing in an RSVP task: an attentional blink? *Journal of experimental psychology* **18**, 849-860 (1992).
8. Vul, E., Nieuwenstein, M. & Kanwisher, N. Temporal selection is suppressed, delayed, and diffused during the attentional blink. *Psychol Sci* **19**, 55-61 (2008).
9. Sergent, C., Baillet, S. & Dehaene, S. Timing of the brain events underlying access to consciousness during the attentional blink. *Nature neuroscience* **8**, 1391-1400 (2005).
10. Sigman, M. & Dehaene, S. Brain mechanisms of serial and parallel processing during dual-task performance. *J Neurosci* **28**, 7585-7598 (2008).
11. Marti, S., King, J.R. & Dehaene, S. Time-Resolved Decoding of Two Processing Chains during Dual-Task Interference. *Neuron* **88**, 1297-1307 (2015).
12. Marti, S., Sigman, M. & Dehaene, S. A shared cortical bottleneck underlying Attentional Blink and Psychological Refractory Period. *NeuroImage* **59**, 2883-2898 (2012).
13. Super, H., Spekreijse, H. & Lamme, V.A. Two distinct modes of sensory processing observed in monkey primary visual cortex (V1). *Nature neuroscience* **4**, 304-310 (2001).

14. Buffalo, E.A., Fries, P., Landman, R., Liang, H. & Desimone, R. A backward progression of attentional effects in the ventral stream. *Proceedings of the National Academy of Sciences of the United States of America* **107**, 361-365 (2010).
15. Hickey, C., van Zoest, W. & Theeuwes, J. The time course of exogenous and endogenous control of covert attention. *Experimental Brain Research* **201**, 789-796 (2010).
16. Schoenfeld, M.A., *et al.* Dynamics of feature binding during object-selective attention. *Proceedings of the National Academy of Sciences of the United States of America* **100**, 11806-11811 (2003).
17. Pomplun, M., Reingold, E.M. & Shen, J. Investigating the visual span in comparative search: the effects of task difficulty and divided attention. *Cognition* **81**, B57-67 (2001).
18. Desikan, R.S., *et al.* An automated labeling system for subdividing the human cerebral cortex on MRI scans into gyral based regions of interest. *NeuroImage* **31**, 968-980 (2006).
19. Olivers, C.N. & Meeter, M. A boost and bounce theory of temporal attention. *Psychological review* **115**, 836-863 (2008).
20. Bowman, H. & Wyble, B. The simultaneous type, serial token model of temporal attention and working memory. *Psychological review* **114**, 38-70 (2007).
21. Peterson, M.S. & Juola, J.F. Evidence for distinct attentional bottlenecks in attention switching and attentional blink tasks. *The Journal of general psychology* **127**, 6-26 (2000).
22. Visser, T.A., Zuvic, S.M., Bischof, W.F. & Di Lollo, V. The attentional blink with targets in different spatial locations. *Psychonomic bulletin & review* **6**, 432-436 (1999).
23. Akyurek, E.G. & Wolff, M.J. Extended temporal integration in rapid serial visual presentation: Attentional control at Lag 1 and beyond. *Acta psychologica* **168**, 50-64 (2016).
24. Jefferies, L.N. & Di Lollo, V. Linear changes in the spatial extent of the focus of attention across time. *Journal of experimental psychology* **35**, 1020-1031 (2009).

Reviewers' comments:

Reviewer #2 (Remarks to the Author):

The authors have done a nice job of addressing the concerns raised in my review and that of the other referee. However, I am still of the opinion, which seems to be shared by the other reviewer, that the manuscript is not of the groundbreaking/novel level of Nature Communications. I have read the author's description of the results and why they are novel, but I would still see this fitting in a more specialist, cognitive neuroscience journal. The strength and limit of the approach used here is that it shows whether there is information in the MEG signal (as measured by a highly sensitive decoding methods) that distinguishes category for the different items for particular sensors at particular times. There are several possible interpretations of these results, but the main message is that the visual system is initially more parallel and then one item is selected. But that is, essentially, what the attentional blink (and many other related phenomena) shows already.

Reviewer #3 (Remarks to the Author):

Discrete and continuous mechanisms of temporal selection in rapid visual streams
Sebastian Marti and Stanislas Dehaene

This is a re-review of this manuscript. In general, the authors have addressed many of my previous concerns and I believe the manuscript is significantly improved. However, I still have some minor concerns:

- 1) The evidence for gradual selection in Figure 4C has changed from the previous version of the manuscript. Why is the effect for T-1 target so much larger than previously?
- 2) My previous specific comment #1 was about why classifiers performed better across time periods than within a time period. I'm not sure my comment was previously clear and therefore, I'm not sure I understand the authors response. For example, it is not clear to me why in Figure 3A a classifier trained at 170 ms could better decode stimulus categories at >400 ms than a classifier trained at that time point. This is also seen in Figure 2B. I understand why a classifier would generalize, but I don't understand why it would do so above what is possible using local data.
- 3) My previous specific comment #3 was in reference to the onset of the 'late' encoding for T-2, T-1, and T stimuli; specifically, that re-activation occurred first for the T-2 stimulus, then the T-1, and finally with the T stimulus. This is highlighted in the authors' reply: they state the second bout of classification occurred at 278 ms, 354 ms, and 410 ms, for T-2, T-1, and T stimuli, respectively. Why?

Response to Reviewer #2:

The authors have done a nice job of addressing the concerns raised in my review and that of the other referee. However, I am still of the opinion, which seems to be shared by the other reviewer, that the manuscript is not of the groundbreaking/novel level of Nature Communications. I have read the author's description of the results and why they are novel, but I would still see this fitting in a more specialist, cognitive neuroscience journal. The strength and limit of the approach used here is that it shows whether there is information in the MEG signal (as measured by a highly sensitive decoding methods) that distinguishes category for the different items for particular sensors at particular times. There are several possible interpretations of these results, but the main message is that the visual system is initially more parallel and then one item is selected. But that is, essentially, what the attentional blink (and many other related phenomena) shows already.

The present study provides a detailed description of how, in the human brain, external information is processed, selected, and gated to awareness. Using a novel combination of magneto-encephalography and multivariate pattern analyses, we show that the mechanisms by which the brain selects external information involves at least two successive operations: a parallel probabilistic selection followed by a serial sampling. To our knowledge, this is the first empirical evidence for the existence of such mechanisms in the brain. This provides new perspectives on how the human brain efficiently processes overwhelming information from the environment.

We clarified the novelty of the study P12, last paragraph.

Response to Reviewer #3:

This is a re-review of this manuscript. In general, the authors have addressed many of my previous concerns and I believe the manuscript is significantly improved. However, I still have some minor concerns:

1) The evidence for gradual selection in Figure 4C has changed from the previous version of the manuscript. Why is the effect for T-1 target so much larger than previously?

Figure 4C shows the average classification performance between 400 and 550 ms after stimulus onset for stimuli at positions T-4 to T+2. Essentially, we found that the classification performance for stimuli at positions T-2, T-1, and T was above chance. In the previous version of the figure, only trials where the target was reported as guess 1 were used. In the current version of the figure, all trials were included because the goal of the figure is to illustrate the effects of the temporal proximity of the target stimulus on nearby distractors. The difference in classification performance between the two versions of the figure is due to the difference in the number of trials included in the analysis. Both versions of this analysis (i.e. with all trials together and trials split according to subjects' report) are fully described in the text P10 §2.

2) My previous specific comment #1 was about why classifiers performed better across time periods than within a time period. I'm not sure my comment was previously clear and therefore, I'm not sure I understand the authors response. For example, it is not clear to me why in Figure 3A a classifier trained at 170 ms could better decode stimulus categories at >400 ms than a classifier trained at that time point. This is also seen in Figure 2B. I understand why a classifier would generalize, but I don't understand why it would do so above what is possible using local data.

The performance of a classifier obviously depends on the quality of the data on which it was trained. A classifier trained at time t and applied at time t' could perform better than a classifier trained and tested at time t' if (i) the two classifiers share similar features and (ii) the signal-to-noise ratio (SNR) is higher at time t than at time t' . To illustrate this point, we generated simulated MEG signals and processed them similarly to the real MEG data. Event-related fields were generated from 100 Gaussian distributions, representing 100 MEG sensors. The mean of each distribution could either be 0 (the sensor is not informative) or .3 (the sensor is informative), resulting in a specific topography. This pattern of activity was generated first between 100 and 300 ms (early time window) and then again between 400 and 600 ms (late time window) but with a lower amplitude (mean of the distribution: .1). This experimental condition was compared to a control condition where only Gaussian noise was present (all sensor means = 0). Fifty trials of 600 ms (50 Hz sampling rate) were generated for each condition. Classifiers were trained to separate the experimental condition from the control condition. The decoding procedure was the same as the one described in the manuscript except that only two classes were used for simplicity purposes.

As expected, the performance of classifiers increased above chance specifically during early and late time window (Figure R1A). When generalized across time, classifiers trained on the early time window performed above chance during both early and late time window. The same pattern was observed for classifiers trained on the late time window. Furthermore, early classifiers – trained on data with better SNR – performed consistently better than late classifiers, even during the late time window. This suggests that in our real MEG data, the classifiers trained at e.g. 170 ms were able to decode stimulus categories at >400 ms with performance even better than classifiers trained at that time point because the data on which those classifiers were trained had a better SNR.

Figure R1. Decoding of simulated MEG recordings. A single pattern of activity was simulated at two different time windows (100-300 ms and 400-600ms). (A) Temporal generalization of classifiers. The x axis corresponds to the testing time, the y axis to the training time. The dotted line represents the diagonal of the matrix. The color code represents the classification performance. (B) Average classification performance as a function of time for classifiers trained in the early (black) or late (red) time window.

3) My previous specific comment #3 was in reference to the onset of the 'late' encoding for T-2, T-1, and T stimuli; specifically, that re-activation occurred first for the T-2 stimulus, then the T-1, and finally with the T stimulus. This is highlighted in the authors' reply: they state the second bout of classification occurred at 278 ms, 354 ms, and 410 ms, for T-2, T-1, and T stimuli, respectively. Why?

First, in order to avoid any confusion, we would like to stress that the second phase of activation for stimuli T-2, T-1 and T were observed at 278 ms, 354 ms, and 410 ms with respect to target onset, which corresponds to 510, 470 and 410 ms with respect to the onset of each stimulus. Second, we can only speculate as to why the effects observed for stimuli at positions T-2 and T-1 preceded the one observed for the target stimulus. One possibility would be that these latencies are influenced by the order of presentation. We proposed in the manuscript that the second phase of activation corresponds to feedback attentional signals (see P10-11). It is possible that these attentional modulations affected first stimulus T-2, then T-1 and finally stimulus T because these were presented one after the other in a sequence, or because stimuli preceding the target were more advanced in the chain of processing. This specific result should be confirmed and further explored in future studies.

We now mention this point P10 §1 in the manuscript.

Reviewer #3 (Remarks to the Author):

The authors have addressed all of my concerns. I especially appreciate the extra simulation analysis for comment #2. My only remaining comment is that I think this is an important point, indicating a difference in SNR between the two epochs, and so I would suggest the authors briefly mention this in the manuscript.

Response to Reviewer #3:

“The authors have addressed all of my concerns. I especially appreciate the extra simulation analysis for comment #2. My only remaining comment is that I think this is an important point, indicating a difference in SNR between the two epochs, and so I would suggest the authors briefly mention this in the manuscript.”

An explanation for the difference in classification performance between classifiers trained at 170 and applied at >400 ms and classifiers trained and tested at >400 ms is now included P7 §1.